# VESPER: global and local cryo-EM map alignment using local density vectors

Xusi Han[1,3], Genki Terashi [1,3], Charles Christoffer [2], Siyang Chen[2] & Daisuke Kihara [1,2✉]

An increasing number of density maps of biological macromolecules have been determined by cryo-electron microscopy (cryo-EM) and stored in the public database, EMDB. To interpret the structural information contained in EM density maps, alignment of maps is an essential step for structure modeling, comparison of maps, and for database search. Here, we developed VESPER, which captures the similarity of underlying molecular structures embedded in density maps by taking local gradient directions into consideration. Compared to existing methods, VESPER achieved substantially more accurate global and local alignment of maps as well as database retrieval.

[1] Department of Biological Sciences, Purdue University, West Lafayette, IN, USA. [2] Department of Computer Science, Purdue University, West Lafayette, IN, USA. [3]These authors contributed equally: Xusi Han, Genki Terashi. ✉email: dkihara@purdue.edu

Proteins are the major working molecules in a variety of cellular processes in a cell. Toward a mechanistic understanding of protein function, tremendous efforts have been paid to solve the three-dimensional (3D) structures of proteins using experimental methods including cryo-electron microscopy (cryo-EM)[1–3], which is increasingly becoming a mainstream technique in structural biology. EM density maps have been rapidly accumulated in the public database, the Electron Microscopy Data Bank (EMDB)[4], which now holds over 12,855 entries as of November 2020. To interpret the structural information contained in EM density maps, alignment of maps is an essential step. Alignment is involved in many important procedures for analyzing EM maps, including rigid-body structure fitting of atomic models[5], comparison of maps to identify differences in maps during a structure modeling process[6] or to understand similarities and differences of different functional states[7–9], and EM map database search[10,11]. Two types of alignment and searching operations, global and local, are valuable for examining EM maps. The former aligns two entire maps to identify corresponding regions of the maps and to identify related maps from a database, while the latter is useful for identifying a substructure captured in a smaller map within the entire complex in a larger map.

There are a couple of existing methods developed for EM map matching and database search. One of them is gmfit, which represents a map with a combination of Gaussian functions (Gaussian mixture model; GMM)[11,12]. gmfit compares two maps by placing one map on the other map in random initial positions, which are then locally optimized by a steepest-descent method to maximize the correlation of the Gaussian functions of the GMM of the two maps. To run gmfit, users need to specify the number of Gaussian distribution functions (GDFs) to approximate a map. A popular molecular visualization program, Chimera, offers a function named fitmap for superimposing two EM maps[13]. To run fitmap, users specify the number of initial placements of maps. Then, from each of the randomly generated placements, local optimization is performed to maximize the correlation between two maps. Previously, we have developed EM-SURFER, a web-based tool for real-time global matching and database search for EM maps[10,14]. In EM-SURFER, we used 3D zernike descriptors (3DZD) for the efficient comparison of EM map isosurfaces. 3DZD is based on a mathematical series expansion of a given 3D function[15,16], which gives a compact and rotation-invariant representation of EM maps. Using 3DZD, an EM-SURFER search against the entire EMDB is completed in a few seconds. On the other hand, EM-SURFER does not provide a map alignment because it uses rotation-invariant descriptors, and it also only performs global matching. Overall, these methods exhibit a limited accuracy in map alignment and a database search, particularly for partial matching. The concept of map alignment has also been used in subunit structure fitting to an EM map of a complex structure, where a simulated density of a subunit is generated and fitted. Such methods include EMFIT[17], FoldHunter[18], ADP_EM[19], and SITUS[20]. EM-LZerD uses a shape-fitting function to select multi-chain complex models that fit to an EM map[21].

Here, we developed a method called VESPER (VEctor-based local SPace ElectRon density map alignment), which performs accurate global and local alignment and comparison of EM maps. VESPER represents an EM map as a set of vectors, which point toward denser points in the vicinity. Thus, the directions of the vectors capture local structures embedded in the map, which turned out to be effective in obtaining accurate global and local map alignment. An alignment of maps is evaluated by a score defined as the sum of dot products of matched vectors from two maps. The best alignment with the maximum score is sought

using a fast Fourier transform (FFT)[22] in an exhaustive search using rotational and translational intervals. Compared on benchmark datasets, VESPER showed a higher accuracy in map retrieval as well as in global and local map matching than gmfit and fitmap.

## Results

We first explain the VESPER algorithm. Then we discuss VESPER's performance in global map matching and partial map matching.

**Overview of the VESPER procedure.** Figure 1a illustrates the overview of the VESPER workflow. For a given density map from EMDB, we used the author-recommended contour level provided in EMDB to extract the occupied volume. An EM map is represented by a set of unit vectors computed with the mean shift algorithm (see the "Methods" section). The voxel spacing of the maps is set to 7 Å. This vector representation originates from the approach used in MAINMAST, a de novo protein structure modeling method for cryo-EM maps[23,24]. In MAINMAST, the locations of density points are updated toward neighboring denser points and clustered iteratively using the mean shift algorithm until the points converge into a small number of representative points. In VESPER, on the other hand, each density point is represented as a unit vector that shows the gradient of the density toward a local representative point that has a high-density value calculated by the mean shift algorithm. In many cases, representative points with high-density values correspond to the backbone of a protein in the map. Thus, the vector representation captures information about underlying local molecular structures around each voxel. For a pair of EM maps, the goal of VESPER is to find the pose transformation that maximizes the agreement of the local density landscape of the two maps. For each rotation of a map using an interval of 30° as default (users can change this interval), a translation scan is performed using FFTs to optimize the sum of dot products of matched vectors (the DOT score). The dot product of a pair of matched vectors ranges from −1 to 1 with 1 for a perfect match, 0 for two perpendicular vectors, and −1 for two vectors pointing in opposite directions. Equal weight is given to the vector at each voxel. A large overlap between two maps tends to have a large DOT score as long as a majority of matched vectors have a positive dot product value. Since the vector has three coordinates, FFT needs to be performed for each component, which makes the time complexity for a DOT score essentially three times larger than other related scores, such as simple cross-correlation that considers one value at each grid point. Then, for each of the 10 top-scoring models from the FFT search, VESPER performs a finer rotational angle search with a 5° interval around each axis (if the angle interval used for the initial coarse-grained search is larger than 5°). Finally, the top 10 (default) or a user-specified number of scoring superimpositions will be output. In the results we show below, we will discuss the top-scoring superimposition for VESPER unless noted otherwise.

On the right-hand side in Fig. 1a, an example of local map alignment with EM maps of a complete V-ATPase structure (EMD-8724) and the $V_o$ region (EMD-8409) is shown. The top panel shows vectors in the two maps (the number of vectors is reduced for illustration). Shown in the bottom panel on the right is the top-scoring superimposition of the two maps, where the map of the $V_o$ region was correctly fitted in the complete map of V-ATPase. The colors of spheres in the superimposed maps indicate the dot product values of matched vectors, with red being a positive score and blue being a negative score or 0. In the magnified region, blue arrows reflect the difference in underlying

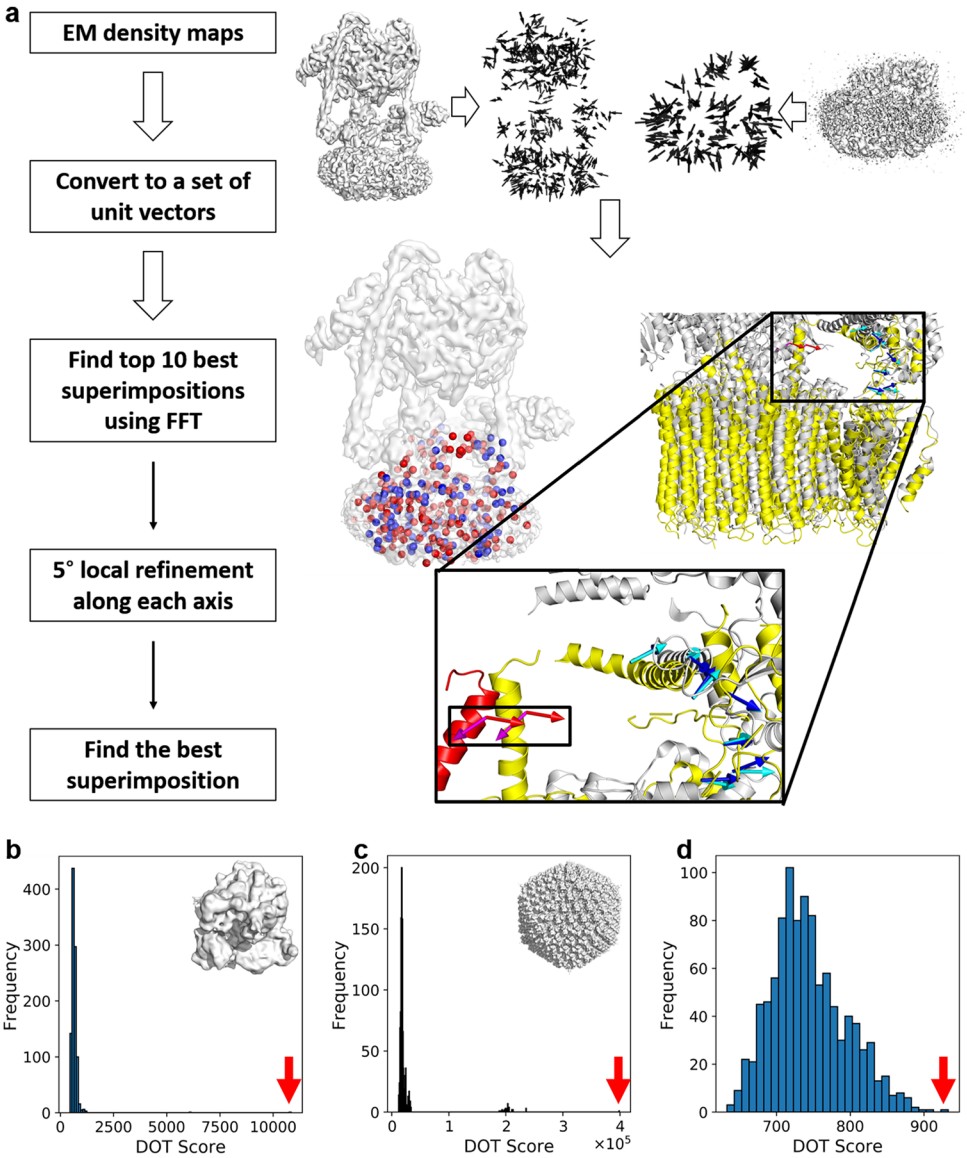

**Fig. 1 Overview of VESPER. a** The flowchart of VESPER. Steps of VESPER are illustrated in the right panel with an example of a map alignment between the complete V-ATPase (EMD-8724, 6.8 Å; left) and V$_o$ region of the V-ATPase (EMD-8409, 3.9 Å; right). First, a set of unit vectors are computed using the mean shift algorithm for each map. The number of vectors for EMD-8724 and EMD-8409 were 2441 and 678, respectively. In the figure, vectors were thinned out for better presentation. Next, the two maps are matched using FFT to maximize the sum of the dot products of matched vectors. Then, the top 10 scoring models undergo a local angle refinement with a 5° interval. The best scoring superimposition is shown at the bottom of the right panel. The V$_o$ region (PDB ID: 5tj5) is colored in yellow and the complete V-ATPase (PDB ID: 5vox) is in gray. In the superimposed maps, vectors with positive and non-positive and DOT scores are colored in blue and red, respectively. The helix in the complete V-ATPase that does not match well to the V$_o$ region is colored in red. Vectors from the V$_o$ region are colored in blue and red, while those from the complete V-ATPase are colored in cyan and magenta. **b** 70S ribosome (EMD-2978, resolution: 11.6 Å) matched to itself. DOT score: 10,841; Z-score: 101.62. **c** human adenovirus 5 capsid (EMD-3004, resolution: 12.5 Å) matched to itself. DOT score, 398,169; Z-score: 94.31. **d** an alignment between human adenovirus (EMD-3004) and 70S ribosome (EMD-2978). DOT score, 943; Z-score: 3.97.

helix orientations (residue 90–115). On the other hand, the helices and loops (residue 304–333) on the right side of the maps have almost identical orientations, which are indicated in red.

**Significance of a map alignment**. For a database search for a query map, we use a normalized score (Z-score) instead of the raw DOT score because the DOT score has a dependency on the size of maps. We compute the Z-score as follows: The query map is placed in a rotational pose with an angle interval used, and for each rotational pose, the query map is translated by the pre-determined translational interval. Then, the largest DOT score

among all the translations for a rotational pose is stored. Examples of the distribution of the largest DOT score from each rotational pose are shown in Fig. 1b–d for self-comparison of 70S ribosome (EMD-2978), self-comparison of human adenovirus 5 capsid (EMD-3004), and a comparison between the two maps, respectively. From each of the top 10 scoring poses found, a further local rotational refinement with an interval of 5° was performed, from which the largest DOT score for two maps was identified. Then, the Z-score for the largest DOT score (indicated with a red arrow in Fig. 1b–d), which is defined as (DOT_Score −mean)/standard_deviation, was computed from the DOT score distribution. Since a DOT score distribution can be biased when

two symmetrical maps, such as virus capsids, are compared, as shown in Fig. 1c, the DOT scores in the distribution are first clustered with single-linkage clustering with a cutoff of 20% of the difference between the maximum and minimum DOT scores, and scores in the largest cluster are used for computing the $Z$-score. For the comparison of virus capsids in Fig. 1c, this clustering process eliminated a bias to the score distribution for the $Z$-score computation, which was introduced from a small peak that locates at around 200,000. This peak came from all rotational poses at the correct translation position of the two maps. The clustering process does not affect usual cases of comparison with asymmetric maps (e.g. Fig. 1b). The $Z$-scores computed for the three map comparisons are 101.62, 94.31, and 3.97 for Fig. 1b–d, respectively. Thus, self-comparisons (Fig. 1b and c) each had a very high $Z$-score, apparently indicating that the compared maps are similar (actually identical as they are self-comparisons) while maps with different shapes have insignificant $Z$-scores (Fig. 1d).

**Dataset of density maps**. To evaluate the performance of VES-PER, we constructed a dataset of EM density maps from EMDB as follows: First, maps that do not have contour level information were excluded. Then, the remaining maps were grouped by the name of the macromolecules of the maps. Groups were inspected manually. A group was removed if it only contains low-resolution maps with a resolution of 20 Å or worse or if it contains fewer than five maps. From each group, five maps were randomly selected. This process yielded a dataset with 129 groups with 645 maps in total. Finally, groups that share the same partial structures are merged into a class. For example, the group for the complete V-ATPase and the group for the $V_o$ domain of V-ATPase were merged in the same class. The resulting dataset with 105 classes was used for evaluating partial map matching performance. The number of maps in a class ranged from 5 to 50.

For global map matching, one group was randomly selected from each class to form a dataset of 82 classes with 410 maps. Thus, each class consists of a single group with five maps. This is to have each class distinct from each other to prevent a query map from having a correct partial match with maps from different but related groups. The list of maps for datasets for global and partial matching are provided in Supplementary Data 1.

Using this dataset, we evaluated a method's ability to retrieve a map in the same class within the top, the first tier, and the second tier. The first tier is defined as the ranks up to the number of other maps in the same class with the query, and the second tier is double of it. Thus, for a global map search, the first and the second tier is within the 4th and the 8th ranks. For a partial map search, the ranks of the first and the second tiers depends on the number of maps in the same class.

**Global map search**. First, we examined the global map search performance of VESPER. We primarily used an angle interval of 30° and a translational interval of 7 Å unless noted otherwise because this setting showed a reasonable balance between the accuracy and the speed among other settings tested (see the "Methods" section). In Fig. 2a–c, we compared results using the DOT score with cross-correlation (CC), which is a commonly used metric to evaluate the fitting of two EM maps[13,25,26]. To compute the DOT score and CC, the density voxel spacing is resampled to 7 Å (or to the user's setting) and maps are contoured before calculation. Both CC and the DOT score were computed for the overlap between the two maps. For a query map in the global map matching dataset, the rest of the maps were compared with the query and ranked by the $Z$-score of the DOT score or CC. We examined if a map in the same group (a correct map) was retrieved as the closest, within the first, or the second

tier. To evaluate map retrieval results for a group, the fraction of query maps in the group that found a correct map within a cutoff rank was computed. The overall performance of a method is computed by the average over all the groups.

Figure 2a shows the histogram of the fraction of maps in each group that found a correct map as the closest hit. As shown, using VESPER with the DOT score (blue bars) had more groups (43 groups) that achieved 1.0 than using CC (orange bars; 28 groups). In Fig. 2b, the retrieval performance of each map group is plotted, considering the first tier. It is apparent that the performance of the DOT score was better than CC for the majority of the map groups. VESPER with the DOT score had a higher correct map fraction for 54 groups, while CC was better for 12 groups. Both methods tied for 16 groups. The same trend was observed when up to the second tier was considered (Supplementary Fig. 1a). The superior performance with the DOT score was observed consistently across all resolution bins from 2 to 50 Å (Fig. 2c and Supplementary Fig. 1b). Figure 2g is an example where VESPER with the DOT score showed a better retrieval performance than CC. For the query map of PKS module 5 (PikAIII) from the pikromycin pathway (EMD-5664[27]), VESPER retrieved all four other maps in the same group, while CC retrieved only one map in the same group.

We further compared the performance of VESPER using the DOT score and CC with three existing methods, gmfit and fitmap, and EM-SURFER that uses 3DZDs for the map shape search[10,14]. gmfit was run with 20 GDFs, which is the setting for map superimposition in the Omokage map search web server where gmfit was used[6]. The other parameters of gmfit were set to their default values except for the maxsize parameter, where we tried two settings: One is default of the standalone program and the other is -maxsize 64, which is the parameter setting used in the Omokage server. In fitmap, an input map was contoured, and the number of initial placements was set to 100. Corr score was used in fitmap. For 3DZD, parameters were set to the same as what are used in EM-SURFER.

Table 1 summarizes the map retrieval performance within the first and the second tier. We added the Laplacian filter in this comparison, which has an effect of enhancing 3D edges[20]. For the global map search (the left half of Table 1), VESPER with the DOT score (VESPER (DOT)) had the best average correct map fractions within the first and the second tier, which was 4.7% points higher than the second-best method, gmfit with the maxsize 64 option and the Laplacian filter, respectively. The direct comparison with gmfit (Fig. 2d) and fitmap (Fig. 2e) shows that VESPER performed better for more map groups in the first tier. The same trend was shown when the second tier was considered (Supplementary Fig. 1c, d).

Figure 2f and Supplementary Fig. 1e show the map retrieval performance for maps at different resolutions. VESPER had the highest fraction of correct maps for most resolution bins. gmfit was the second for most of the resolution bins and the best for the resolution bin of 12–14 Å. Figure 2h is an example of map search from a query map of the ClpB–ClpP complex (EMD-2558[28], resolution: 21 Å) where VESPER performed better than gmfit in map retrieval. While VESPER found all the other four maps of the same complex in the first tier, gmfit retrieved two unrelated maps that have somewhat similar overall shape at the third and fourth ranks, which happened perhaps due to the low resolution of the query map. Figure 2i is the opposite case, where VESPER's retrieval result was worse than gmfit. gmfit's retrievals were all correct in the first tier for the query map of the secretin GspD (EMD-6675[29]) while VESPER's third and the fourth retrievals were incorrect, both from GroEL. For this query, the GroEL maps had relatively high score because they have an overall similar shape and also because these maps are largely hollow inside, and thus inconsistency inside the maps were not much penalized.

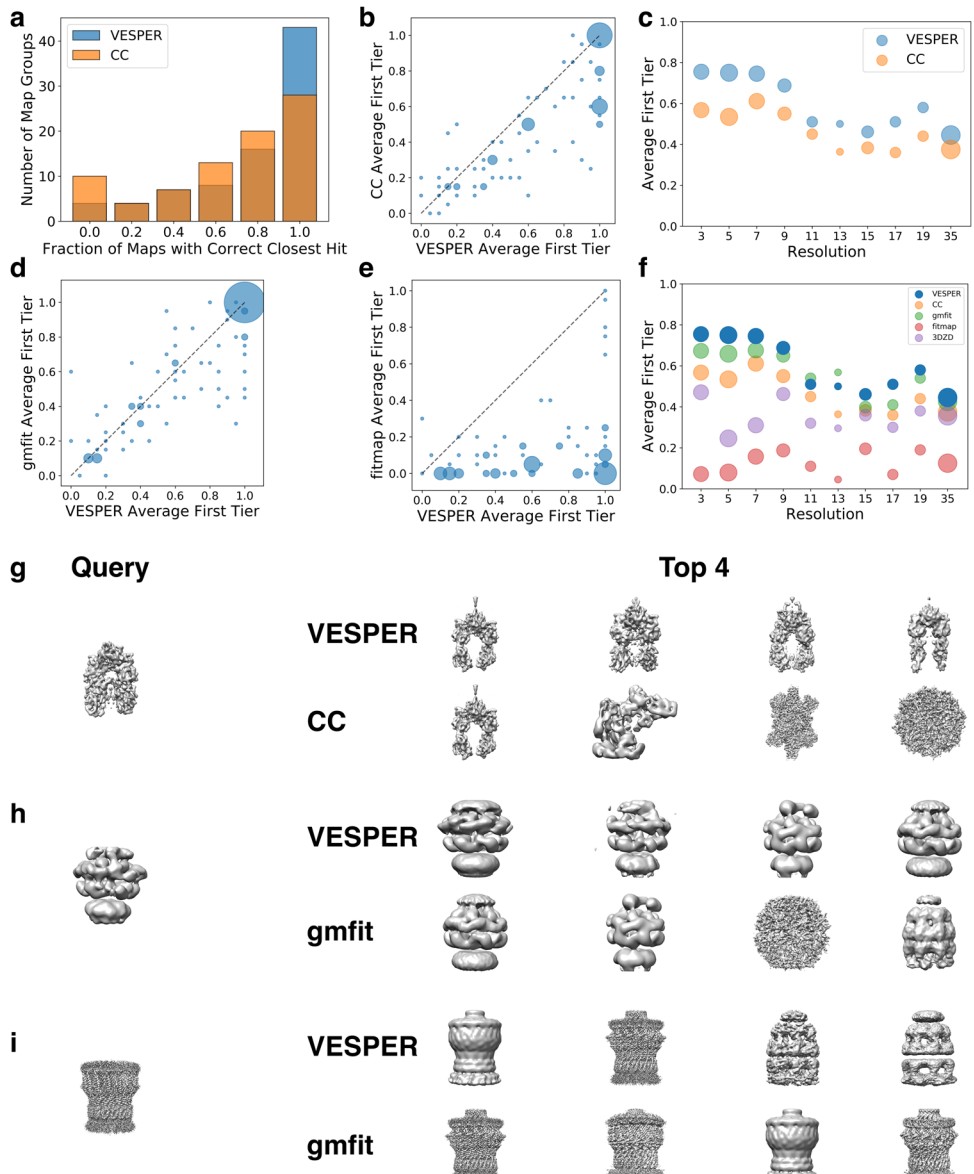

**Fig. 2 Performance on global map search. a–c** comparison of VESPER with the DOT (blue) score and CC (orange). **a** The number of map groups classified by the fraction of member maps that retrieved a correct map as the top hit. **b** The first tier hit fraction for each group. The area of a data point is proportional to the number of groups. **c** The average fraction of correct hits within the first tier on maps at different resolutions. **d** Comparison between VESPER and gmfit on the first tier hit fraction. **e** Comparison between VESPER and fitmap on the first tier hit fraction. **f** The average first tier hit fraction for maps in each resolution bin by VESPER with the DOT score (blue), CC (orange), gmfit (green), fitmap (red), and EM-SURFER (3DZD; purple). The resolution of the query map was considered. **g** An example of a query map where the DOT score performed better than CC. The query map is PikAIII (EMD-5664). The top four retrieved maps by VESPER with the DOT score were all from PikAIII: EMD-5649, EMD-5663, EMD-5651, and EMD-5666, in this order. On the other hand, only 1 out of the top 4 retrieved maps by CC were PikAIII: EMD-5649 (PikAIII), EMD-6443 (*Tetrahymena* telomerase), EMD-6635 (glutamate dehydrogenase), EMD-5145 (bovine TriC), in this order. **h** An example of map retrieval where VESPER performed better than gmfit. The query is a map of ClpB bound to ClpP (EMD-2558). All the four maps retrieved in the first tier by VESPER were ClpB-ClpP: EMD-2557, EMD-2556, EMD-2560, EMD-2559 in this order. With gmfit, only two within the top four retrieved maps were the ClpB-ClpP: EMD-2559 (ClpB-ClpP), EMD-2560 (ClpB-ClpP), EMD-5145 (bovine TriC), EMD-2327 (GroEL-GroES). **i** An example of map retrieval where gmfit performed better than VESPER. The query is a 3.04 Å res. map of secretin GspD (EMD-6675). VESPER retrieved only two correct maps among the top four retrieved maps: EMD-1763 (secretin GspD), EMD-6676 (secretin GspD), EMD-2325 (GroEL-GroES), and EMD-1203 (GroEL-gp31) in this order. All four retrieved maps by gmfit were all from secretin GspD: EMD-6676, EMD-8779, EMD-1763, and EMD-6677 in this order.

**Global map alignment accuracy.** Next, we examined the global map alignment accuracy. For this test, we randomly selected three pairs of maps, each from the same resolution range of better than 5 Å, 5–10 Å, and over 10 Å, and six maps pairs that were taken from different resolution ranges. These maps have a fitted protein structure in PDB (Table 2). The ground truth of the superimposition for the map pairs was computed by aligning the underlying protein structures of the maps using MM-align[30]. Table 2 summarizes the root mean square deviation (RMSD) of the best-scoring superimposition by VESPER with the DOT score

**Table 1 Average fraction of correct maps retrieved within the first and the second tier.**

|  | Global | | Partial | |
|---|---|---|---|---|
|  | FT | ST | FT | ST |
| VESPER (DOT) | **0.613** | **0.670** | **0.592** | **0.657** |
| CC | 0.479 | 0.551 | 0.456 | 0.515 |
| Laplacian | 0.564 | 0.623 | 0.510 | 0.567 |
| gmfit | 0.563 | 0.609 | 0.479 | 0.551 |
| gmfit (−64)* | 0.566 | 0.613 | 0.479 | 0.556 |
| fitmap | 0.124 | 0.164 | 0.101 | 0.123 |
| EM-SURFER | 0.350 | 0.398 | 0.285 | 0.339 |

Global and partial map search results are shown. FT, the fraction of correct maps within the first tier (top|C|−1 maps, where |C| represents the number of maps in the same class as the query map); ST, the fraction in the second tier (top 2*(|C|−1) maps). VESPER (DOT) is the results of VESPER using the DOT score. For the Laplacian filter, cross correlation was computed after the filter was applied to maps. gmfit used the default parameter of the gmfit program. gmfit(−64) used a "-maxsize 64" option when converting an EM map to a Gaussian mixture model using the gmconvert program. This is the parameter used in the gmfit webserver. The number in bold shows the best performance for each metric.

in comparison with CC, gmfit, and fitmap. For VESPER, four parameter combinations of a voxel spacing and a rotational angle were examined.

For 10 out of 15 map pairs, VESPER showed the lowest RMSD using one of the parameters used among the methods compared (Table 2). Comparing VESPER with the DOT score and CC, we see that the DOT score had more cases with a smaller RMSD, indicating that the DOT score performs better than scoring with CC, which is consistent with the global map search accuracy discussed with Fig. 1 and Table 1. The difference of RMSD values between VESPER with the DOT score and CC for the 15 maps has a $p$-value of 0.008 when tested with one-sided paired $t$-test. Examining VESPER's results for maps with less (better) than 5 Å resolution, the RMSDs achieved became lower (better) as finer voxel spacings were used. This implies that the DOT score was able to distinguish small differences in the alignments. The same trend still held, but was less obvious, for maps with worse resolutions. In Supplementary Fig. 2 we provided figures of alignments with large RMSD values by fitmap and CC. VESPER performed better than gmfit and fitmap. fitmap produced alignments with large RMSD values for a few cases. This may be because fitmap performs local optimization from random initial structures. Overall, VEPSER showed the best performance among the methods compared for both global map database search and alignment.

**Comparison with 18 existing map alignment scores**. Additionally, we compared the DOT score with 18 existing scores for evaluating map alignments, which were described in the paper by Joseph et al. [8] The scores were compared on a dataset of 100 map alignments for 26 EM map pairs that were used in the paper. The reference alignment of a map pair was computed by superimposition of the underlying protein structures. Alignments were classified to correct (more precisely, sufficiently close to the reference alignment) and incorrect using either RMSD or the log (ALCPS) score. To comprehensively evaluate the performance of the scores, we used seven evaluation metrics, which capture different important aspects of the performance of the scores for identifying correct alignments among other candidates. For more details, refer to the "Methods" section and Supplementary Information file. The results are provided in Supplementary Data 2.

The results (Supplementary Data 2) show that, overall, the DOT score performed the best among the scores compared.

**Table 2 Global map alignment by VESPER (DOT), CC, gmfit, and fitmap.**

| Res. range | Map 1 IDs | Map 2 IDs | PDB RMSD (Å) | RMSD (Å) | | | | | | | | | gm-fit | fit-map |
|---|---|---|---|---|---|---|---|---|---|---|---|---|---|---|
| | | | | 1 Å, 10° | | 3 Å, 10° | | 5 Å, 10° | | 7 Å, 30° | | | |
| | | | | VES | CC | VES | CC | VES | CC | VES | CC | | |
| <5 Å | 3240/5fn5 | 2677/5a63 | 1.91 | **2.21** | 2.38 | 3.61 | 3.92 | 3.25 | 3.25 | 8.87 | 8.87 | 2.63 | 2.90 |
| | 8881/5wpq | 8764/5w3s | 2.08 | 1.12 | **1.12** | 2.05 | 2.05 | 2.05 | 2.05 | 2.05 | 2.05 | 1.19 | 56.99 |
| | 9515/5gjw | 6475/3jbr | 4.37 | **2.31** | 3.12 | 2.88 | 6.65 | 5.51 | 4.18 | 5.76 | 5.76 | 2.95 | 97.48 |
| 5–10 Å | 8744/5vy8 | 8267/5kne | 3.44 | **0.86** | **0.86** | 1.74 | 2.93 | 2.01 | 2.01 | 2.01 | 2.01 | 2.30 | 73.67 |
| | 6284/3j9t | 8724/5vox | 5.13 | 2.79 | 2.67 | 3.67 | 2.89 | 3.67 | 3.67 | 3.67 | 3.67 | 5.05 | **1.04** |
| | 3342/5fwm | 3341/5fwl | 1.45 | **0.56** | **0.56** | **0.56** | **0.56** | **0.56** | **0.56** | **0.56** | **0.56** | 3.60 | 4.98 |
| >10 Å | 1961/4a0v | 1962/4a0w | 10.44 | 4.74 | 5.39 | 4.94 | 6.11 | 4.94 | 4.94 | 4.94 | 4.94 | 8.17 | 7.61 |
| | 2557/4d2u | 2559/4d2x | 6.96 | 3.27 | **2.85** | 3.95 | 3.02 | 6.09 | 6.09 | 8.64 | 8.64 | 7.38 | 44.31 |
| | 4547/6qg5 | 4548/6qg6 | 4.51 | **1.00** | 1.78 | **1.00** | 5.34 | **1.00** | **1.00** | **1.00** | **1.00** | 5.21 | 1.63 |
| Cross-res. | 8784/5w9l (3.6) | 8789/5w9n (5.0) | 8.33 | **2.84** | **2.84** | **2.84** | **2.84** | **2.84** | **2.84** | **2.84** | **2.84** | 4.69 | 79.51 |
| | 9515/5gjw (3.9) | 6476/3jbr (6.1) | 4.37 | **3.12** | 3.34 | 6.86 | 8.13 | 5.47 | 5.47 | 4.17 | 8.27 | 6.06 | 64.19 |
| | 3238/5fn3 (4.1) | 2678/5a63 (5.4) | 0.68 | 3.34 | 3.34 | 3.80 | 3.80 | 3.93 | 3.93 | 8.25 | 8.55 | **3.22** | 3.68 |
| | 2000/4aas (8.5) | 5001/3cau (4.2) | 7.88 | **5.85** | 81.98 | 8.45 | 130.77 | 10.02 | 78.46 | 7.16 | 77.71 | 6.72 | 6.35 |
| | 8673/5vh9 (7.7) | 8706/5vlj (10.5) | 6.24 | 3.15 | 4.31 | 3.11 | 44.67 | 3.68 | 41.26 | 5.73 | 6.08 | 4.49 | **1.71** |
| | 2325/3zpz (8.9) | 2327/3zql (15.9) | 2.09 | 3.91 | 4.37 | 4.71 | 4.71 | 7.19 | 7.40 | 9.62 | 9.81 | 4.97 | **0.37** |
| Average | | | 4.66 | 2.74 | 8.06 | 3.61 | 15.23 | 4.04 | 11.14 | 5.02 | 10.05 | 4.58 | 29.76 |
| ΔAvg. | | | - | −5.32 | | −11.61 | | −7.10 | | −5.03 | | - | - |

Res. Range, resolution range. The last six pairs of in the cross-resolution category are pairs that are from different resolution ranges. Map 1(2) IDs columns list EMDB and PDB IDs of the two maps and the associated PDB entries. For the cross-resolution pairs, the map resolution is shown in the parentheses. The PDB RMSD column provides RMSD values of the PDB files computed with MMalign. The RMSD columns of each alignment method show the deviation of the top-scoring alignments from the golden-standard alignment computed by MMalign. When computing RMSD values, equivalent alignments due to symmetry were taken into account. For VESPER (VES) and Cross-correlation (CC), four different shifting and angle interval combinations were used. The average row shows the average value of each column. ΔAvg. shows the average RMSD difference, VES(PER)−CC for the same voxel and angle spacing setting. The smallest RMSD value for each map pair is highlighted in bold. In this comparison, 3D-SURFER was not used as it does not provide map alignment.

Other scores that showed the top performance in terms of some metrics include a normal vector-based score (NVA in the table), the overlap (OVR), and the cross correlation (CCC).

**Partial map search**. Next, we discuss VESPER's performance in partial map search. The partial map search is aimed at finding maps in a dataset, which contains common macromolecules with the query map. The results are summarized in the right half of Table 1 and Fig. 3. Table 1 shows that VESPER (DOT) achieved the highest average success rates when retrievals within the first

and the second tier were considered. When the top hit was considered, VESPER (DOT) had more map groups (67 groups) than CC (51 groups) that had a 100% successful retrieval (Fig. 3a). When individual map groups were considered, VESPER (DOT) was more accurate than CC for the majority of map groups (Fig. 3b), and VESPER's advantage was consistent over all the resolution ranges (Fig. 3c). When compared to gmfit (Fig. 3d) and fitmap (Fig. 3e), it was clear that VESPER performed better in the map retrieval for more map groups. Comparison for maps determined at different resolutions (Fig. 3f) shows that VESPER

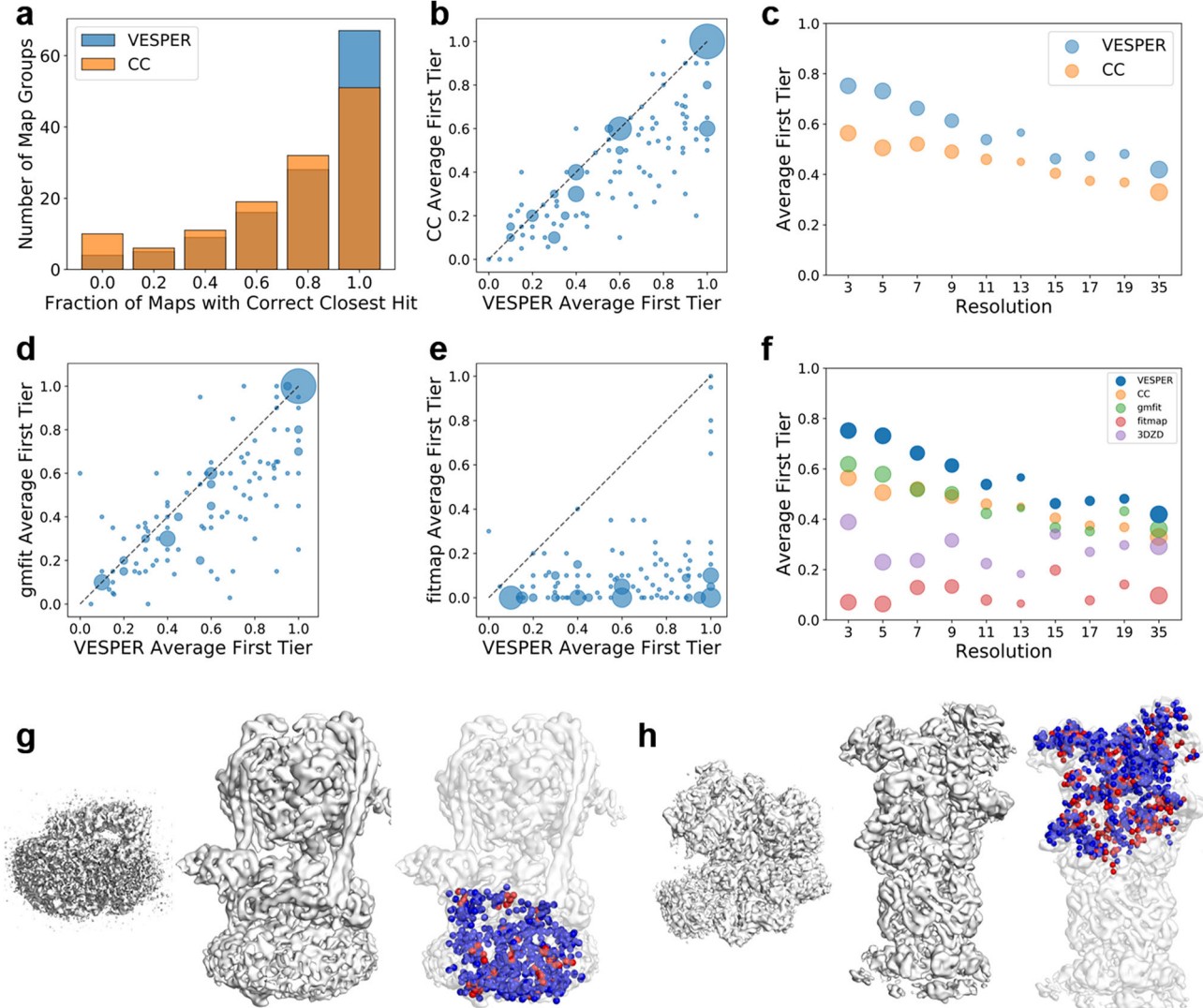

**Fig. 3 Performance on partial map search. a** The number of map groups with different fractions of member maps with a correct top hit. VESPER with the DOT score (blue) and CC (orange). **b** The average fraction of correct hits within the first tier for the 129 groups. The x-axis, VESPER with the DOT score; the y-axis, CC. The area of a point is proportional to the number of groups at that data point. **c** Comparison of VESPER (DOT) and CC on partial map retrieval at different resolutions. The resolution of the query map was considered. The average fraction of correct hits within the first tier was considered. **d** Comparison between VESPER and gmfit on the average fraction of correct hits in partial map search within the first tier for each map group. **e** Comparison between VESPER and fitmap on the first tier hit fraction in partial map search. **f** The average first tier hit fraction for maps in each resolution bin for VESPER (DOT) (blue), CC (orange), gmfit (green), fitmap (red), and EM-SURFER (3DZD; purple). The resolution of the query map was considered on the x-axis. **g** The $V_o$ domain of V-ATPase (left, EMD-8409, res.: 3.9 Å) matched to the complete V-ATPase (middle, EMD-8726, res.: 7.6 Å). Colored dots in the right panel shows the dot product of matched vectors, with blue being a positive score and red for zero or a negative score. For this query, the first tier success rates of VESPER (DOT)/CC/gmfit/fitmap were 0.57/0.36/0.36/0.21, respectively. The ranks of this hit (EMD-8726) from the query by VESPER (DOT)/CC/gmfit/fitmap were 3/524/66/67 and the RMSD values of the match computed with the underlying protein subunit were 6.05/132.45/140.23/2.27 Å, respectively. **h** proteasome regulatory particle (left, EMD-8675, res.: 6.1 Å) matched to 26S proteasome (middle, EMD-3537, res.: 7.7 Å). The first tier success rates of VESPER (DOT)/CC/gmfit/fitmap were 0.89/0.32/0.37/0.11, respectively. The ranks of this hit (EMD-8726) from the query by VESPER (DOT)/CC/gmfit/fitmap were 1/507/184/473 and the RMSD values of the match computed with the underlying protein subunit were 11.32/111.40/138.33/131.44 Å, respectively.

was the best for all resolution bins. The same trend was observed when different criteria were used for evaluation (Supplementary Fig. 3).

Two examples of local matches are shown in Fig. 3g, h. The first example is a search from a map of the $V_o$ domain of V-ATPase (EMD-8409), which found a map of the complete V-ATPase (EMD-8726) at the third rank (Fig. 3g). This retrieval at the high rank contributed to a substantially higher first-tier (FT) success rate by VESPER in comparison with the other methods. VESPER's FT success rate with the DOT score was 0.57, while CC, gmfit, and fitmap only achieved 0.36, 0.36, and 0.21, respectively. As shown in the right panel of Fig. 3g, the majorities of vectors from the two maps have positive dot product score (i.e. they point to similar orientations), which yielded a high retrieval rank. The RMSD of the local alignment by VESPER (DOT) was 6.05 Å, which was sufficient to capture the map similarity. The local matches by CC and gmfit were not successful as shown as very large RMSD values of the alignments of 132.45 and 140.23 Å. In terms of RMSD, fitmap had a better RMSD for this map pair, 2.27 Å, but this match was ranked as low as 67 in the search. The second example (Fig. 3h) is from a search from the proteasome regulatory particle (EMD-8675), which is aligned with 26S proteasome (EMD-3537). Although the alignment was not highly accurate (an RMSD of 11.32 Å), it was sufficient to rank the full 26S proteasome map as the top rank in the search. The other three methods had a completely wrong alignment with an RMSD over 100 Å and could not retrieve this map within a high rank (see the figure caption).

In Fig. 4, we asked a question whether a search tends to retrieve maps in the same group higher in the rank than maps in the same class. The answer was that the result depended on each case. Out of 100 query maps, a map in the same group (yellow) was ranked at the top of the retrieval for 46 cases while the top was from the same class (green) for other 50 cases. When the retrieved maps up to third rank for each query was counted (thus in total of 300 maps), there were 125 maps in the same group and 158 maps from the same class. Thus, about half of the top ranked retrievals are from the same group and the other half were from the same class.

**Atomic model fitting accuracy.** In the last section, we discuss atomic model fitting accuracy of the methods. Nine maps were selected for a test set, three each from resolution ranges of better than 5, 5–10 Å, and over 10 Å (Table 3). These maps each have an associated PDB entry of a protein structure that covers most of the region of the maps and does not contain nucleic acid structures. For each protein chain, a density map is simulated using the molmap command in Chimera at the resolution of the target map. The density threshold was set to 0.2. With VESPER, fitting was computed with four parameter combinations of a voxel spacing and an angle interval as performed in Table 2.

The results are summarized in Fig. 5. In this figure, VESPER with the DOT score and with CC was run with a voxel spacing of 3 Å and a rotational angle of 10°. Results with other parameter combinations are shown in Supplementary Fig. 4. The first panel, Fig. 5a, shows the RMSD values of the top-scoring alignment of 57 queries computed by the four methods. The plot compares VESPER (DOT) with each of the other methods. Among the 57 queries, VESPER (DOT) placed more protein chains, 30 (52.6%) and 33 (57.9%) within an RMSD of 5.0 and 10.0 Å, than the other three methods, as shown in the plot. CC, gmfit, and fitmap had 15/15, 10/12, 14/16, maps within an RMSD of 5.0/10.0 Å, respectively. The difference of the performance between VESPER (DOT) and CC was statistically significant with a $p$-value < 0.05 computed with the one-sided paired $t$-test. Figure 5b is a

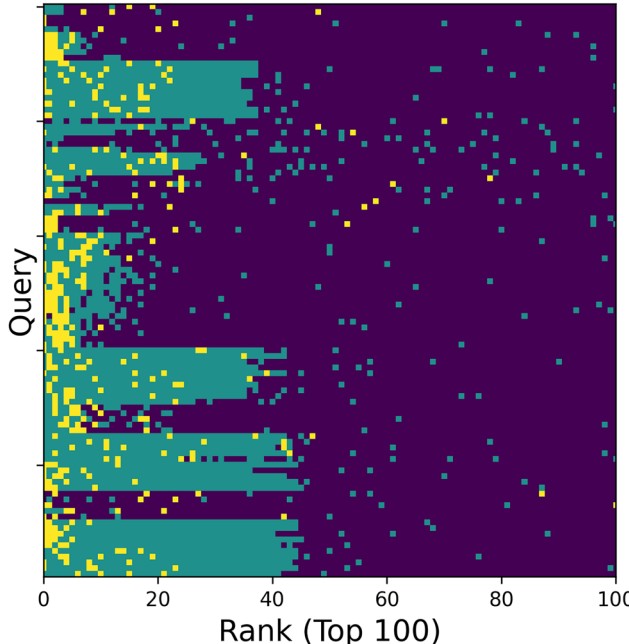

**Fig. 4 Top 100 retrievals of partial map search.** For 100 queries of partial map search that belong to a class with 10 or more members, top 100 retrieved maps were visualized in colors indicating whether the maps belong to the same group (yellow), the same class (green), or others (dark blue).

breakdown of the alignment results for each EM map in Table 3. It counted the fraction of query chains that were aligned within 5.0 and 10.0 Å by each method. Considering the 5.0 Å cutoff, VESPER (DOT) showed the largest fraction of correctly placed chains for seven out of nine maps including two ties with other methods.

The next three panels show examples of chain structure fitting by the four methods. The first example (Fig. 5c) is results for five chains of γ-secretase (EMD-3238, PDB ID: 5fn3). VESPER (DOT) successfully fit four chains within an RMSD of 5.0 Å to the correct pose (red). In contrast, fitmap placed two chains, nicastrin (chain A) and PEN-2 (chain D), and CC and gmfit placed only nicastrin within 5.0 Å RMSD. Similar results were observed in the next example, transcription factor IIH (EMD-3802, PDB ID: 5of4) (Fig. 5d). Out of 10 chains, VESPER (DOT) (red) placed eight chains (chain A, B, E, F, H, X, Y, Z) closer than an RMSD of 5.0 Å. CC and gmfit placed three chains and one chain, respectively, within the cutoff, fitmap did not find a pose within 5.0 Å RMSD for any chain. In the panel, the pose of subunit XPD (chain B) computed by fitmap is shown, which has an RMSD of 37.7 Å but still the smallest RMSD among the other chains. The chain was placed in the correct position in the map but with a substantial rotation from the correct orientation, which made the RMSD large. The last panel (Fig. 5e) shows fitting for 18 chains of RNA polymerase I-Rrn3–CF complex (EMD-3591, PDB ID: 5n5z). VESPER (DOT) placed seven chains within 5.0 Å RMSD, which was substantially more than the other three methods. gmfit did not fit a chain within 5.0 Å but one chain at a position with an RMSD of 9.0 Å.

**Discussion**

In this work, we developed VESPER, a method for EM density map search and alignment. The VESPER algorithm matches two maps by considering local gradients represented by unit vectors, which captures underlying macromolecular structures in the

**Table 3 EM maps used in partial map alignment evaluation.**

| Resolution bins | Protein name | Map ID | Number of chains | Chain IDs |
|---|---|---|---|---|
| <5 Å | Voltage-gated calcium channel | 9515/5gjw | 4 | A, C, E, F |
| | γ-secretase | 3238/5fn3 | 5 | A–D, G |
| | Transcription factor IIH[a] | 3802/5of4 | 10 | A, B, D–H, X–Z |
| 5–10 Å | Voltage-gated calcium channel | 6476/3jbr | 4 | A, B, E, F |
| | RNA polymerase I-Rrn3-CF complex[a] | 3591/5n5z | 18 | A–R |
| | Hsp90-Cdc37-Cdk4 complex[a] | 3340/5fwp | 4 | A, B, E, K |
| >10 Å | Dynein-Lis1 complex | 8706/5vlj | 3 | A–C |
| | NMDA receptor | 8104/5ipt | 4 | A–D |
| | Origin recognition complex[a] | 8541/5ujm | 5 | A–E |

Map ID shows the EMD-ID and the associated PDB ID of the macromolecules. The Chain IDs show the chains that were used as queries of the local alignment evaluation and the number of chains indicates the number of query chains.
[a]The names of the four entries that were shown in Fig. 5.

maps. This implementation of the DOT score often results in different map alignments as compared with CC. In CC, positions with large absolute density values, such as those in a high-density region in a map, influence more to the overall CC value. On the other hand, for the DOT score the contribution of each aligned position pair is essentially the same because the vectors are normalized to the same length. But this also means that the DOT score can be affected by changes in local gradient caused by small structure variations.

Overall, VESPER showed higher accuracy than existing methods in both global and local EM map search and alignment under a reasonable speed requirement and for a given range of resolutions under the parameter settings tested. Note that, in general, the optimal parameter setting for a method differs for each map and the purpose of the computation. Thus, a perfectly fair comparison is not possible, and the comparison shown in this work is to characterize the performance of VESPER but not to rank the methods.

With VEPSER, an accurate map database search can be provided, for example, to EMDB, which currently does not offer on-the-fly context-based map search. Another useful application of VESPER is to perform local map alignment to identify a subunit in a density map of a macromolecular complex. Since VESPER outputs multiple candidate alignments with a fitness score, users can manually examine alternative alignments and choose the most plausible one considering background information of the complex. By identifying the location of known subunit structures in an EM map, VESPER maybe also helpful for segmenting the map. Overall, we expect VESPER will serve as an indispensable addition to the structural biology toolbox for studying EM maps.

## Methods

**Unit vector representation of local densities in VESPER.** VESPER represents a density map with a unit vector at each grid position, which points toward a neighboring local density maximum point. The mean shift algorithm, a non-parametric clustering approach, is used for this task. First, the grid spacing of a map is converted to 7 Å. Then, a unit vector is placed at each grid point $\mathbf{x}_i$ ($i = 1, \ldots, N$) with a density value that is no less than the author-recommended contour level $\Phi_{\mathrm{thr}}$ in an EM map. The unit vector located at $\mathbf{x}_i$ is $\rightarrow \frac{(\mathbf{y}_i - \mathbf{x}_i)}{|\mathbf{y}_i - \mathbf{x}_i|}$, where $\mathbf{y}_i$ is computed as follows:

$$\mathbf{y}_i = \frac{\sum_{n=1}^{N} k(\mathbf{x}_i - \mathbf{x}_n) \Phi(\mathbf{x}_n) \mathbf{x}_n}{\sum_{n'=1}^{N} k(\mathbf{x}_i - \mathbf{x}_{n'}) \Phi(\mathbf{x}_{n'})} \quad (1)$$

$k(p)$ is a Gaussian kernel function, which is defined as

$$k(p) = \exp\left(-1.5 \left|\frac{p}{\sigma}\right|^2\right), \quad (2)$$

where the $\sigma$ is a bandwidth, which was set to 8.0 in all the computations in this work. $\Phi(\mathbf{x}_n)$ is the density value of the grid point $\mathbf{x}_n$. The Gaussian kernel has an effect of reducing density noise.

**Cross-correlation (CC).** CC for an alignment of two maps is computed in the same way as in other existing software[31–34]:

$$\mathrm{CC} = \frac{\sum_{i=1}^{N} (u_i - \bar{u})(v_i - \bar{v})}{\sqrt{\sum (u_j - \bar{u})^2} \sqrt{\sum (v_j - \bar{v})^2}}, \quad (3)$$

where $u_i$ and $v_i$ are a density value of the position $i$ in two maps, $\bar{u}$ and $\bar{v}$ are the average density value of the two maps within the contour level used. $N$ in the numerator is the number of overlapped grid points in the alignment.

**Exploration of parameter combinations.** The voxel spacing and the angle spacing are the two parameters for using VESPER. We used a voxel spacing of 7 Å and an angle spacing of 30° for aligning and searching density maps as the default setting of VESPER we used in this study. This setting was chosen from several parameter combinations we examined because it provided a reasonable balance between the map search accuracy and the computational time. In Table 4 and Supplementary Fig. 5, we provided the computational time and the global map retrieval accuracy of parameter combinations with a voxel spacing of 3, 7, or 10 Å and an angle spacing of 10°, 30°, 60°, and 90°. Results shown are the average of three query maps (EMD-3661, EMD-8724, and EMD-1203) against all the 410 maps in the global matching dataset.

The computational time increased about 6 to over 30 times when the voxel spacing was changed from 7 to 3 Å while it showed a relatively smaller decrease to about when a larger spacing of 10 Å was used. Using a finer spacing of 10° also increased the computational cost about 3–20 times from 30°. Comparing the time needed for using the DOT score and CC, CC costs about the half the time of DOT score when an angle spacing of 10° was used, but the differences became smaller when less expensive settings were used.

Turning our attention to the retrieval accuracy (Supplementary Fig. 5), 10° and 30° did not make a substantial difference, but using coarser-grained angles, such as 60° or 90°, drastically deteriorated the accuracy. The voxel spacing of 7 Å was also practically convenient because it is the grid spacing of the density maps we used. To further speed up a search, practically we could apply a pre-filtering to reduce the number of maps in the database to search against. For example, maps that have a significantly different volume to the query may be removed. We could also remove some functional classes of maps, e.g. virus entries or ribosome entries, or maps of a certain resolution range, if the user is not interested in them.

In addition to the voxel and angle spacing, the contour level to use for extracting input maps would affect the accuracy. In this work, we used author-recommended contour level provided in EMDB for each map.

**Recommended Z-core cutoff value.** For the global and partial map search, a Z-score of 10 or larger is an indication of significant map similarity, judging from Z-score distributions of map pairs of positive (maps of the same group for global map search and maps in the same class for local map search) and negative cases (Supplementary Fig. 6). For map alignment, a Z-score of 10 would also be a proper cutoff as shown in the distribution of the local map alignments (i.e. local structure fitting). Below a Z-score of 10, results are mixture between positive and negative search results and alignments better or worse than a 5 Å RMSD.

**Comparison of the DOT score with 18 existing scores.** We followed the paper by Joseph et al.[8] to compare the DOT score with 18 other existing scores in their performance of selecting accurate map alignments. 18 scores are: Overlap (OVR); Segment based Manders' Overlap Coefficient, Local cross correlation (SMOC); Local cross correlation (SCCC); Cross correlation-coefficient (CCC); Local mutual information (LMI); Normalized mutual information (NMI); Chamfer surface distance score on points selected based on a density threshold range (CDT); Chamfer surface distance score on points selected using mean filter (CDM); Chamfer surface distance score on all points at an iso-contour level (CDA); Normal vector score on surface points selected from a density threshold range (NVT); Normal vector score

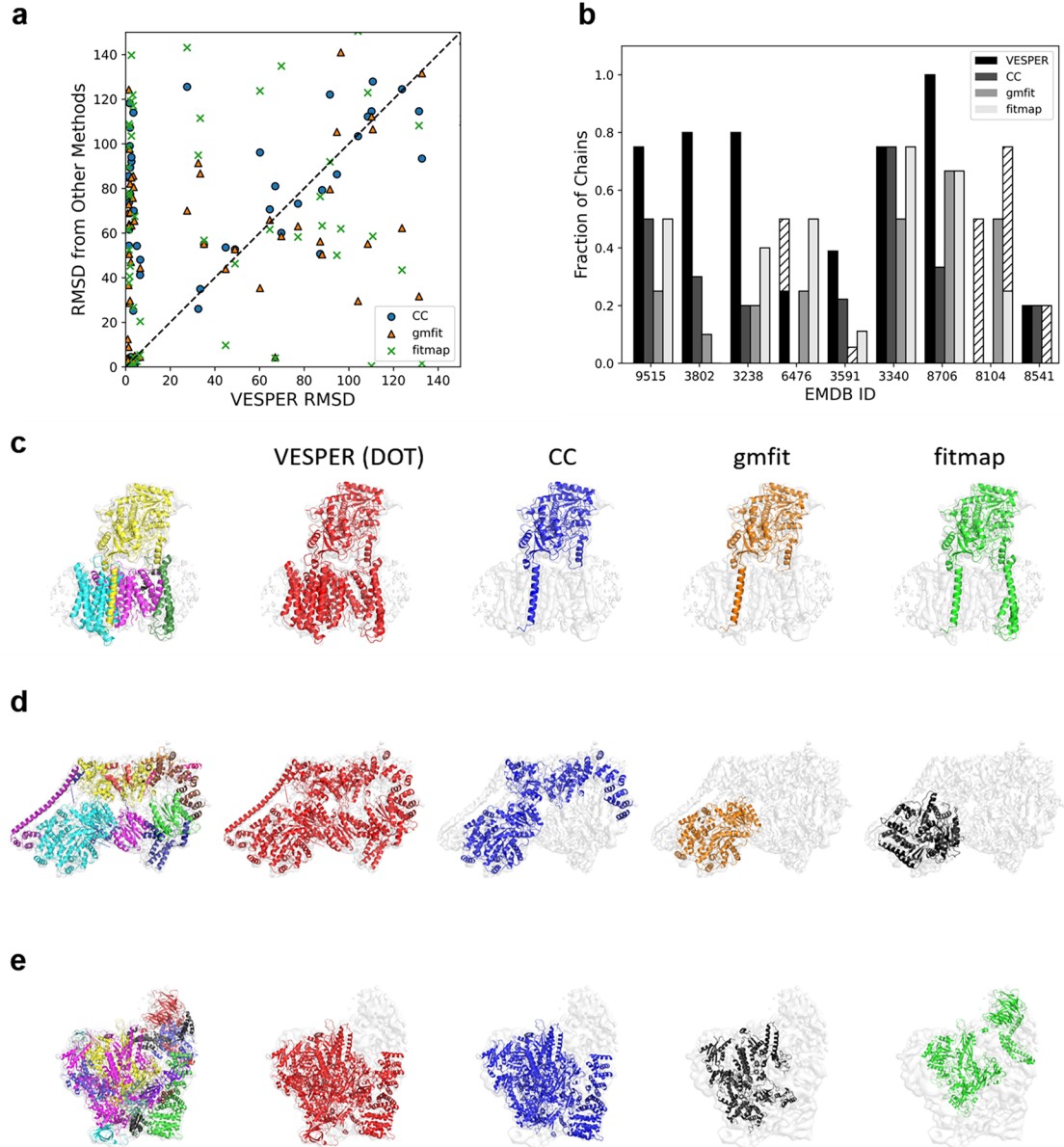

**Fig. 5 Performance of atomic model fitting. a** Comparison of RMSD of the alignment with the best score by VESPER (DOT) with CC, gmfit, and fitmap. EM maps in the dataset are listed in Table 3. A voxel spacing of 3 Å and a rotation angle of 10° were used in Fig. 4. Supplementary Fig. 4 provides results for the other three parameter combinations. Blue circles, comparison against CC; orange triangles, gmfit; green crosses, fitmap, respectively. **b** The fraction of query chains for each map that had the top-scoring alignment with an RMSD of 5.0 Å or less (solid gray bars) and 10.0 Å or less (including hatched bars). Black bars, VESPER (DOT); dark gray, CC; medium gray, gmfit; pale gray, fitmap. **c** Chain structure fitting for five chains of γ-secretase (EMD-3238, PDB ID: 5fn3). The left panel shows superimposed structures of 5fn3 in the map. Each chain is in different color. From the second to the last panel from the left, chains placed within an RMSD of 5.0 Å are shown for VESPER (DOT), CC, gmfit, and fitmap, respectively. The number of chains placed within the cutoff was 4, 1, 1, and 2 chains by these methods, respectively. **d** Chain structure fitting of transcription factor IIH (EMD-3802, PDB ID: 5of4). There are 10 chains to fit. VESPER (DOT), CC, gmfit, and fitmap placed eight (A, B, E, F, H, X, Y, Z), three (A, B, Y), one (B), and zero chains within 5.0 Å RMSD, respectively. Chain IDs are taken from the PDB file. For fitmap, the placement of chain B, which had an RMSD of 37.7 Å is shown, since this chain had the smallest RMSD among the other chains. **e** fitting of 18 chains of RNA polymerase I-Rrn3-CF complex (EMD-3591, PDB ID: 5n5z). Within 5.0 Å RMSD, VESPER (DOT), CC, gmfit, and fitmap placed seven (A, B, C, E, G, H, O), four (A, B, C, O), zero, and two (B, P) chains, respectively. For gmfit, chain A that was fit at an RMSD of 9.0 Å is shown.

on surface points identified by mean filter on binary mask (NVM); Normal vector score on all points at an iso-contour level (NVA); Normalized variants of CDT, CDM, & CDA (CDT_GDT), (CDM_GDT), (CDA_GDT); SMOC, SCCC, LMI are combined with OVR (SMOC_OV), (SCCC_OV), (LMI_OV). These scores were computed with the TEMPy program[35].

The alignment dataset was computed for 26 map pairs provided in the Table 1 of the paper by Joseph et al. Two map pairs from the table were excluded from our dataset because defining the correct alignment was not possible since the associated atomic detailed structures from PDB do not sufficiently overlap with the density maps.

All maps were resampled to the grid spacing of 3 Å. 100 alignments were computed for each pair of maps by shifting and rotating maps using the translation and angle definition used in the ALCPS score[8]. ALCPS is defined as $ALCPS = 2\pi r\theta/360$, where $r$ is the translation and $\theta$ is a rotation angle from the correct superimposition, which was defined by the superimposition of the associated protein structures using MM-Align[30]. For an alignment set for a map pair, correct alignments were defined in two metrics, RMSD and log(ALPCS). When RMSD was used for evaluation, 10 Å was used as the cutoff to define correct alignments. For log(ALPCS), cutoffs were defined differently for the three categories, which were −0.4, 0.82, and −0.5, for the Other, Ribosome,

**Table 4 CPU hours for combinations of voxel and angle spacing settings.**

|  | Voxel Spacing (Å) | | |
| --- | --- | --- | --- |
|  | **3** | **7** | **10** |
| *DOT score* | | | |
| Comp. Vector | 8.09 (0.020) | 1.21 (0.0029) | 0.76 (0.0019) |
| Angle 10° | 5643.2 (13.8) | 158.9 (0.388) | 44.5 (0.109) |
| Angle 30° | 250.8 (0.612) | 18.94 (0.0462) | 8.42 (0.0205) |
| Angle 60° | 129.9 (0.317) | 13.39 (0.0327) | 7.12 (0.0174) |
| Angle 90° | 120.4 (0.294) | 12.80 (0.0312) | 6.93 (0.0169) |
| *CC* | | | |
| G. kernel | 7.42 (0.0181) | 1.15 (0.0028) | 0.74 (0.0018) |
| Angle 10° | 2962.9 (7.227) | 96.1 (0.234) | 24.61 (0.060) |
| Angle 30° | 197.5 (0.482) | 14.6 (0.0355) | 7.15 (0.0174) |
| Angle 60° | 98.0 (0.239) | 11.5 (0.0280) | 6.49 (0.0158) |
| Angle 90° | 93.3 (0.227) | 11.2 (0.0274) | 2.82 (0.0069) |

For each voxel spacing and angle spacing combination, the average CPU hours by VESPER on global map search for three query maps, EMD-3661, EMD-8724, and EMD-1203, against the global map dataset of 410 maps. The upper half of the table shows times for using the DOT score while the latter half shows the times for using CC. Comp. Vector in the DOT score category shows the computational time needed for computing vector representation of the 410 maps. G. kernel in the CC category shows the time needed for applying the Gaussian kernel (Eq. (2)) to the 410 maps. The values shown are an average for processing the 410 maps and 410 comparisons. The times for preparing one file or comparing a pair of maps are shown in parentheses. gmfit took 72.7 CPU hours for computing the Gaussian mixture models and search by gmfit took 0.09 CPU hours. Fitmap took 14.7 CPU hours. 3DZD took 0.2 CPU hours, among which the 3DZD computation took almost all the time. We used Intel Xeon E5 processor@2.60 GHz with 128 GB memory on the Halstead cluster computer at Purdue to measure the times.

and Virus categories, respectively. These log(ALCPS) cutoff values were adopted from the aforementioned paper by Joseph et al.[8].

Scores were evaluated by six metrics, the area under the curve (AUC) of the receiver operator characteristic (ROC) curve, average precision (i.e. the area under the precision-recall curve), the Z-score of the score of the best alignment (i.e. the alignment that is the closest to the reference alignment), the average Z-score of top 10 best alignments, Δlog(ALCPS) and ΔRMSD, which are the difference of the best alignment and the top-choice by the score, the average accuracy, and the average F1-score. A Z-score is defined by (score−average_of_the_score)/standard deviation, where the average and the standard deviation were computed from the score distribution of different alignments of the same map pair. An F1-score is defined as 2∗(precision∗recall)/(precision + recall). These scores evaluate different important aspects of the scores. log(ALCPS) considers deviations of translational and rotational shifts of an alignment, whereas RMSD measures an average deviation at each amino acid residue point. Supplementary Fig. 7 provides correlation of RMSD with the DOT Z-score and four other representative scores. AUC and the average precision are for evaluating the overall retrieval performance, the two Z-score metrics evaluate how distinctively a score can select correct alignments, Δlog (ALCPS) and ΔRMSD are for evaluating how good the top choice of a score is relative to the best alignment in the dataset, and the average accuracy and F1-score check the classification performance of a score. Since the suitability of each metric differs for different purposes and targets, we provide all these values in Supplementary Data 2 (in a separate Excel file). Refer also to explanation in the Supplementary Information file for some more details. The alignment dataset is made available at https://kiharalab.org/vesper_data. The VESPER program[36] is available at https://github.com/kiharalab/VESPER.

## Data availability
The dataset of EM maps is provided in Supplementary Data 1. The experimental EM maps can be downloaded from EMDB. The datasets used for the comparison with 18 existing scores and the local structure alignment (Fig. 5) are provided at https://kiharalab.org/vesper_data.

## Code availability
The VESPER program is freely available for academic use via https://github.com/kiharalab/VESPER and https://kiharalab.org/em-surfer/vesper.php.

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

## Acknowledgements

This work was partly supported by the National Institutes of Health (R01GM133840, R01GM123055), the National Science Foundation (DMS1614777, CMMI1825941, and MCB1925643) and the Purdue Institute of Drug Discovery.

## Author contributions

D.K. conceived the study. G.T. developed the algorithm and coded VESPER. X.H. built the dataset, optimized the score, and carried out most of the computations. X.H., G.T., C.C., and D.K. analyzed the data. G.T. and C.C. compared the DOT score with 18 existing scores. G.T. performed atomic model fitting. S.C. built the VESPER web server and G.T. built the data resource webpage. X.H. drafted and D.K. edited and finalized the manuscript.

## Competing interests
The authors declare no competing interests.
