## [Peer Review File · Nature Communications]

REVIEWER COMMENTS

Reviewer #1 (Remarks to the Author):

This paper proposed a new computer program VESPER, which aligns two density maps determined by electron microscopy. The authors proposed a new score (VESPER DOT score) using a vector to neighbor representative points determined by the mean-shift algorithm, although they employed a standard searching algorithm, based on FFT. After the revolution of Cryo EM technology around 2013, many density maps of near atomic resolution have been determined and stored in EMDB, and for most of these high-resolution maps, atomic models have been built and stored in PDB. If the corresponding atomic models are available, model-to-model alignment is more accurate. And model-to-map alignment is necessary for modeling process. This paper deals with map-to-map alignment and searching maps from the query map, however, map-to-map comparison is not popularly used by EM researchers. The authors have to clearly address why map-to-map alignment is valuable to publish this paper in Nature communications.

I also have following major requests before publication mainly about performance comparison of their method with other methods.

(1) As I pointed out, usefulness of map-to-map alignment and searching is not obvious for most of the readers. The authors have to address how map-to-map alignment works for the research and show biological findings by map-to-map alignments reported recently.

(2) The originality of this study is the VESPER DOT score, not its searching algorithm. Therefore, I request the authors to compare more other scores except cross-correlation using their searching program. The meaning of VESPER score is quite difficult to understand, but I guess it is a kind of edge (or surface) enhancing filter for low-resolution map by combination of smoothing and gradient. Chacon and Wriggers (2002) stated Laplacian-filter is useful to align low-resolution map and implemented as the colores program in SITUS.

Chacón P, Wriggers W. Multi-resolution contour-based fitting of macromolecular structures. *J Mol Biol.* 2002 Mar 29;317(3):375-84.

I strongly recommend to implement Laplacian filter in their algorithm, and compare its performance. I guess the implementation of the Laplacian-map in their FFT program is not so difficult. And I also recommend to test dot product of simple gradient vector without using mean-shift representative points.

(3) Joseph et al. (2017) tested alignment accuracies of 18 different scores and reported "We find that the performance of different scores vary with the map-type, resolution and the extent of overlap between volumes" and "A combined score involving the local mutual information and overlap (LMI_OV) performs best overall, irrespective of the map category, resolution or the extent of overlap, and we recommend this score for general use". The authors just concluded "DOT score performs better than scoring with CC" for alignment accuracy, but they have to discuss and compare about other scores mentioned in Joseph et al. (2017) in Introduction or Discussion sections.

Joseph AP, Lagerstedt I, Patwardhan A, Topf M, Winn M. Improved metrics for comparing structures of macromolecular assemblies determined by 3D electron-microscopy. *J Struct Biol.* 2017 Jul;199(1):12-26. doi: 10.1016/j.jsb.2017.05.007. Epub 2017 May 25.

(4) The authors do not discuss computational times and resources about the methods. They have to show how much computational costs are required for their methods and compare them with other methods. I also want to know costs for making the mean-shift representative points.

(5) For the “Global map search”, which scores were used for gmfit and fitmap ? In particular, I wonder why “fitmap” performed so bad as shown in Fig 2f. The “fitmap” function outputs three scores: Corr, Ave and Inside. Which score among the three was used for “fitmap” ? I also wonder the gmfit was improperly executed. I tried several map pairs using their Web service of gmfit (<https://pdbj.org/gmfit/pairgmfit.html>) against the query map EMD-2558. Obtained CC is as follows : 0.719 for 2559, 0.762 for 2560, 0.641 for 5145, 0.596 for 2327, 0.862 for 2557 and 0.736 for 2556. This rank is inconsistent with Fig 2h. Why ? Did author use other score except CC for gmfit ?

(6) In page 14, the authors wrote “Comparing VESPER and CC, we see that VESPER had more cases with a smaller RMSD, indicating that the DOT score performs better than scoring with CC”, but, I do not think so. I feel Table 2 shows VES and CC generate very similar RMSDs, not indicating DOT is better than CC. In order to convince suspicious readers, the authors have to perform a rigorous statistical test if they want to say DOT is better than CC.

(7) In Table 2, the authors should add RMSD values calculated by fitting of atomic model by MMalign. It is because if the two conformations are more different, the fitting of corresponding maps becomes more difficult. RMSD of MMalign should be ideal minimum value for fitting.

(8) In Table 2, some complexes have symmetries; TRPML ch has C4, ClpB-ClpP has C7 symmetry. Did the author care their symmetries when they calculated RMSD? The reason why I asked about question is, fitmap often returns very large RMSD (52-97 angstrom). Is it due to the symmetry problem?

(9) In Table 2, the pair EMD-9758 and EMD-9778 is not good for bench mark, because the corresponding atomic models 6j0a and 6j45 only cover very small parts of the maps, they cannot be used as the correct standard for the fitting. The authors should choose another map pair “randomly” again.

(10) In the “Partial map alignment accuracy” section, a density map for each chain of atomic model was manually generated with the help of “Zone” function of UCSF Chimera. I think this test is not well-designed due to following two reasons. First, such a small density map is rarely obtained by current single particle analysis. Second, because the benchmark data was generated by the authors’ hands, other researchers cannot reproduce these maps to evaluate their upcoming new methods. To rescue these problems, I recommend to perform to fit a chain-based atomic model on the map, instead of a chain-based density map on the map. Because an atomic model is converted to the map for fitting (like colores in SITUS), the authors can do it by subtle modification of their source code. If the authors really do not want to perform model-to-map alignment, the authors have to open their hand-making bench-mark data in their WEB site or somewhere.

(11) In Eq.2, the authors’ kernel is defined as $\exp(-1.5 |p/\sigma|^2)$. The standard Gaussian function is defined as $\exp(-0.5 |p/\sigma|^2)$. Do they have a special reason to employ 1.5 instead of 0.5 ?

(12) I guess two parameters are important for representing maps: bandwidth sigma (=8

angstrom) and voxel spacing (= 7 angstrom). However, the authors employed smaller voxel spacing (1, 3, and 5 angstrom) in Table 2 (Global alignment). When the voxel spacing =1, 3, 5 angstrom, did they employ smaller bandwidth less than 8 angstrom? How much was voxel spacing for the “Partial map alignment accuracy” calculation?

(13) For a typical map, how many number of mean points are generated? Please add number of mean points for the case of EMD-8409 and EMD-8724 in the legend of Fig.1.

Reviewer #2 (Remarks to the Author):

In this manuscript, the authors describe a new method to quickly align EM maps by first generating a set of gradient vectors, and aligning vectors in two maps at a time by conversion to Fourier space.

Overall the paper is written well, the method is described clearly, and the results are well-organized. It is a novel and innovative method than others in the field, mainly through the use of gradient vectors for alignment purposes. However, gradients are known to be easily disrupted by noise, which is typically the case in cryoEM maps, and it is not clear if this was thoroughly addressed.

I think such a paper is timely in the field and the method could be useful, however I do have concerns with the limited accuracy at high resolutions. The advantage to other methods, e.g. fitmap, which can be tuned to have a wider search, or exhaustive search methods, other than speed alone, seems limited. Some parts may also need a bit more clarification as detailed below.

Introduction

- It is stated that “an accurate algorithm for aligning, comparing, and searching maps would substantially increase the value of the accumulated maps in EMDB”. It may not be too clear how this is so, and some points to elucidate some use cases could be interesting to add. It is typically more common to match an atomic structure (e.g. from the PDB) to a map, so as to search new maps for known molecular components. Perhaps this may be a point where such a method would be useful, i.e. to match an existing map with known protein components to the new map. If this method only works between maps, a model could be first turned into a map and used for search, making it more useful.
- Several existing methods are mentioned, such as gmfit, fitmap. Some other commonly used ones like Situs, ADP_EM, phenix.place_model_in_map, and Foldhunter are not mentioned. These are a lot more thorough than fitmap and would offer a more challenging comparison. Some may only place models, but at least ADP_EM and Foldhunter should work with map-map as well. I would say such comparisons may be not be necessary to perform for publication because it would add extra work, but they would make the paper a lot more informative and potentially relevant – i.e. a very important question is: is this method as good or close to an exhaustive search, which even though computationally expensive, is the only way to make sure good potential fits are not missed?

Results

- The voxel spacing of 7Å seems extremely large. This map should have an original voxel spacing of ~1Å or so. Were other voxel spacing tried and how do they affect the results?

This likely means that maps are only matched at low resolutions, but what does this mean for high-resolution maps?

- Using the Z-score is a good way to go. This has been used before rather than pure match score to show that fits are good.
- Figure 2. It is not clear if the text on page 11 is a caption or the main text.
- It is not too clear how the CC score is used. Sounds like it is still with the VESPER search, but in that case, it's not the VESPER search that is important, but rather the use of the DOT score. In that case, the statement 'VESPER had a better retrieval performance than CC' seems ambiguous.
- Comparison to other methods (gmfit, fitmap, 3DZD). The results are well presented and quite thorough. However the other methods could potentially perform just as well if they are given different parameters (e.g. to increase their search space). It is not too clear how the parameters were chosen, other than to make running times similar.
- Partial map alignment accuracy: as stated, all methods performed rather bad at 5.0Å and higher (note: the phrase "relatively higher resolution of less than 5.0 Å" may be stated a bit better). Could a more thorough search with voxel spacing of less than 7Å in the VESPER search help?
- It is a bit discouraging that the method would fail at higher resolution, where more detail should allow for higher accuracy. This may come back to the score used rather than the method, and exhaustive search should be used. This seems to say that VESPER is not comparable to exhaustive search unfortunately, but if speed is needed rather than accuracy, it would have the advantage.

Discussion

- Perhaps VESPER may not be described as an 'accurate method' as per the results above, especially at higher resolutions. Perhaps this can be stated more in a way to indicate that it has higher accuracy under certain speed requirements and for a given range of resolutions.

Methods

- It would be interesting if the parameters for the other method used were given and discussed briefly. E.g. for the fitmap method, what was N? Fitmap uses random placements so different runs could give different results?

Some final thoughts:

- the use of local gradients may mean that noise is amplified. How can this be avoided? Does the reduction to voxel size of 7Å use appropriate smoothing or just binning? If smoothing is used, this may help with reducing the effect of noise.
- The use of a threshold would ideally be avoided. This means that if the user specifies a different threshold, then the method would have different results. Also, all maps could be considered even if a threshold is not given.

Review by Grigore Pintilie

Reviewer #3 (Remarks to the Author):

With the rapidly increasing number of cryoEM structures determined, tools for comparison of this data is very relevant and useful for the community. The proposed approach uses vector representation of volumes based on mean shift algorithm, used in a previously published method for protein structure modelling from EM volumes. The authors report improvement

over other related approaches for retrieval of related structures from a dataset. Although the proposed tool looks promising, the strengths (and weaknesses) have to be demonstrated better with more examples and benchmarks. The computation time is a limitation as well for database searches. I have a few major concerns to do with demonstration of strengths/weaknesses of the proposed approach and examples to help user interpret results.

- 1) The computational time is a very important factor for database searches. More accurate search retrieval requires finer rotational sampling (10^o, Table 2, Table S2). This table is important from a user's point of view and has to be moved to the main text. The computational time can perhaps be reduced with volume pre-processing into vector representations? Strategies have to be discussed.
- 2) The approach is compared against other related methods for volume comparison and alignment. However, the parameters used for other methods have to be defined clearly. For example:
 - a) For CC: Were the volumes resampled into 7A voxels?, and maps contoured before score calculation? Was the calculation limited to volume of overlap between the two maps? CC is quite sensitive to these factors.
 - b) how was the number of gaussians decided for running gmfit? Omokage server retrieves slightly different results to gmfit. For example: search with EMD-2558 seems to retrieve 2557 as top hit, and 2556 and 2560, before EMD-5145 and EMD-2327 (Fig 2h).
 - c) For fitmap, were the maps pre-processed and contoured? Which metric was used for alignment?
- 3) Comparison of maps at different resolutions: In the tests for global and local accuracy, the alignments were tested within resolution categories. Is there a reason for not testing across the resolution ranges? This has to be demonstrated and discussed. Also for the case of comparison of a range of different conformations, it is useful to demonstrate this with a few examples.
- 4) For the global searches, I would be interested in knowing how the performance is when multiple groups of a single class are included in the search dataset, whether the partial alignments are ranked lower to full alignments?
- 5) For VESPER, which parameters need user optimization? Do you recommend a Z-score cut-off for searches and alignments?
- 6) Figure 1 has to be improved. Would be more intuitive to indicate positive DOT score positions with blue and negative with red. Also the atomic model segments of V-ATPase and Vo can be shown in the context of full model superposition.

Responses to Comments by Reviewer #1:

This paper proposed a new computer program VESPER, which aligns two density maps determined by electron microscopy. The authors proposed a new score (VESPER DOT score) using a vector to neighbor representative points determined by the mean-shift algorithm, although they employed a standard searching algorithm, based on FFT.

After the revolution of Cryo EM technology around 2013, many density maps of near atomic resolution have been determined and stored in EMDB, and for most of these high-resolution maps, atomic models have been built and stored in PDB. If the corresponding atomic models are available, model-to-model alignment is more accurate. And model-to-map alignment is necessary for modeling process. This paper deals with map-to-map alignment and searching maps from the query map, however, map-to-map comparison is not popularly used by EM researchers. The authors have to clearly address why map-to-map alignment is valuable to publish this paper in Nature communications.

I also have following major requests before publication mainly about performance comparison of their method with other methods.

(1) As I pointed out, usefulness of map-to-map alignment and searching is not obvious for most of the readers. The authors have to address how map-to-map alignment works for the research and show biological findings by map-to-map alignments reported recently.

I agree that the sentence was not well written. We have rewritten the sentence in Introduction and Abstract. The sentences now read:

(Abstract): To interpret the structural information contained in EM density maps, alignment of maps is an essential step for structure modeling, comparison of maps, and for database search.

(Introduction): To interpret the structural information contained in EM density maps, alignment of maps is an essential step. Alignment is involved in many important procedures for analyzing EM maps, including rigid-body structure fitting of atomic models⁵, comparison of maps to identify differences in maps during a structure modeling process⁶ or to understand similarities and differences of different functional states⁷⁻⁹, and EM map database search^{10,11}.

Relevant works are cited in the sentence in Introduction.

Related to this, I would like to mention that, we now completely replaced the partial map alignment results with new results of atomic structure model rigid-body fitting in order to

(2) The originality of this study is the VESPER DOT score, not its searching algorithm. Therefore, I request the authors to compare more other scores except cross-correlation using their searching program. The meaning of VESPER score is quite difficult to understand, but I guess it is a kind of edge (or surface) enhancing filter for low-resolution map by combination of smoothing and gradient. Chacon and Wriggers (2002) stated Laplacian-filter is useful to align low-resolution map and implemented as the colores program in SITUS.

Chacón P, Wriggers W. Multi-resolution contour-based fitting of macromolecular structures. *J Mol Biol.* 2002 Mar 29;317(3):375-84.

I strongly recommend to implement Laplacian filter in their algorithm, and compare its performance. I guess the implementation of the Laplacian-map in their FFT program is not so difficult. And I also recommend to test dot product of simple gradient vector without using mean-shift representative points.

Thank you for the good suggestion. We have implemented the Laplacian filter in the VESPER code and tested the performance in the global and local map search as shown in the revised Table 1. As shown in the table, the Laplacian filter did perform better than CC and gmfit in all the results other than the first tier (FT), but VESPER still performed the best.

We cited the paper by Chacon & Wriggers (2002), which explains the Laplacian filter.

The VESPER code with the Laplacian filter implementation is now made available at the github page. <http://github.com/kiharalab/VESPER>.

Regarding the VESPER algorithm, it is not the edge enhancing filter. From each density point, a gradient vector is computed. The vector indicates, basically, the direction from the voxel to the neighboring high dense point, which often corresponds to the main-chain of the proteins in the map. And this vector is computed to all the points in the map, not only the mean-shift representative points. Therefore, it is already doing exactly what you suggested. To clarify the algorithm, we added more explanation in page 6.

(3) Joseph et al. (2017) tested alignment accuracies of 18 different scores and reported “We find that the performance of different scores vary with the map-type, resolution and the extent of overlap between volumes” and “A combined score involving the local mutual information and overlap (LMI_OV) performs best overall, irrespective of the map category, resolution or the extent of overlap, and we recommend this score for general use”. The authors just concluded “DOT score performs better than scoring with CC” for alignment accuracy, but they have to discuss and compare about other scores mentioned in Joseph et al. (2017) in Introduction or Discussion sections.

Joseph AP, Lagerstedt I, Patwardhan A, Topf M, Winn M. Improved metrics for comparing structures of macromolecular assemblies determined by 3D electron-microscopy. *J Struct Biol.* 2017 Jul;199(1):12-26. doi: 10.1016/j.jsb.2017.05.007. Epub 2017 May 25.

Thank you for the interesting suggestion. We have newly performed comparison with the 18 scores that were reported in the above-mentioned paper by Joseph et al. All the results are provided in the new Supplementary Table 2 in a separate Excel file. The procedures and the evaluation metrics are explained in the Supplementary Information (a PDF file) in page 2-3 and in the Method section in the main manuscript in page 26-27. The results are described in the page 16.

Here we explain some more about the experiments: The dataset of EM map pairs was taken from Table 1 of the paper by Joseph et al. but two EM map pairs were excluded because the PDB entries associated to the maps do not overlap sufficiently with the maps and were not able to compute the reference map alignment. In the paper by Joseph et al., the distributions of the alignments were generated by running gmfit. However, we found that using gmfit was problematic for making the alignment distribution for many EM map pairs, because gmfit could not sample correct alignments that satisfy the $\log(\text{ALCPS})$ cutoff, and as the result, the distribution only contained only one correct alignment, which is the reference (i.e. correct) alignment itself. This is shown in Figure 7 of their paper (for example, the plot at the right bottom of Figure 7 only contains a single data point at $\log(\text{ALCPS}) = -2$, which is, as the figure caption states, the reference alignment itself). This problem also happened when we ran gmfit by ourselves. Therefore, instead of using gmfit, we generated the alignment distribution by sampling the translation and angle space explicitly. We generated 100 alignments for each map pair. The procedure is described in the supplementary information file and in the manuscript.

The dataset of the alignments is provided for readers to use at http://kiharalab.org/vesper_data.

Second, in addition to using $\log(\text{ALCPS})$ as the metric to judge the accuracy of a map alignment, we also used RMSD of the underlined protein structures of the two maps to classify an alignment into “correct (sufficiently close to the reference alignment)” or “incorrect”. This is because we found that $\log(\text{ALCPS})$ can be very small and thus an alignment can be considered as “correct” even if two maps do not seem aligned well when the alignment is examined on Chimera. This happens because ALCPS is defined as $2\pi r\theta/360$, a multiplication of translation r and rotation angle θ , and thus a combination of a small r (e.g. 0.01) and a large θ (e.g. 30+ degree) can be considered to be correct using the $\log(\text{ALCPS})$ cutoff values used in their paper. We think RMSD is a better metric than $\log(\text{ALCPS})$ because it does not have the ambiguity that occurs with $\log(\text{ALCPS})$. But we evaluated alignments with both $\log(\text{ALCPS})$ and RMSD separately, which are shown in separate pages in the Excel file of the Supplementary Table 2.

Third, in addition to the AUC value, we used five more evaluation metric of each score, average precision, the z-score of the score of the correct alignments (z-score 1 and z-score 10), $\Delta[\log(\text{ALCPS})]$ and $\Delta(\text{RMSD})$, which is the difference of the best alignment and the top-choice by the score, the average accuracy, and the average F1-score. These scores are commonly used in protein structure prediction and give a comprehensive view of performance of a computational method (, which is here, the 18 scores and the DOT score). These are described in Methods.

As shown in Supplementary Table 2, when RMSD was used (the first page of the Excel file), clearly the DOT score performs the best among the all the scores. When $\log(\text{ALCPS})$ was used (the second page), the DOT score was best for all three categories (other, ribosome, virus) when z-score 10, the best accuracy, and the best F1-score were considered, and best in ribosome and

virus categories when z-score 1 was considered. CCC, OVR, CDA, and NVA was the best in terms of some metrics for some categories.

In Supplementary Table 2, the best value for a metric (each column) is highlighted in bold.

(4) The authors do not discuss computational times and resources about the methods. They have to show how much computational costs are required for their methods and compare them with other methods. I also want to know costs for making the mean-shift representative points.

In Table 3 (this is the originally Supplementary Table 2. We moved it to the main text as requested by reviewer 3), we provided the computational time for gmfit, fitmap, and 3DZD in table caption. We also provided the time cost for using mean-shift algorithms to compute the vector representation of points.

(5) For the “Global map search”, which scores were used for gmfit and fitmap ? In particular, I wonder why “fitmap” performed so bad as shown in Fig 2f. The “fitmap” function outputs three scores: Corr, Ave and Inside. Which score among the three was used for “fitmap” ? I also wonder the gmfit was improperly executed. I tried several map pairs using their Web service of gmfit (<https://pdbj.org/gmfit/pairgmfit.html>) against the query map EMD-2558. Obtained CC is as follows : 0.719 for 2559, 0.762 for 2560, 0.641 for 5145, 0.596 for 2327, 0.862 for 2557 and 0.736 for 2556. This rank is inconsistent with Fig 2h. Why ? Did author use other score except CC for gmfit ?

For fitmap, Corr was used. We clarified it in the text (page 12).

Regarding to the question about gmfit, the difference observed by the reviewer was due to an option used in gmfit: In the Omokage (or gmfit) server, an EM map is converted to gaussian mixture models by the command “gmconvert”. The server uses the gmconvert with an option “-maxsize 64”. This option is not a default parameter of gmfit, which we used, and it caused the different results. Below in the table, we performed gmconvert with -maxsize 64 option. Then re-computed gmfit with emd-2558 as a query. Overall, the results of running “gmfit with (gmconvert -maxsize 64)” on our local machine was consistent with the server:

Query:
2558

EMID	gmfit with (gmconvert -maxsize 64)	gmfit	Gmfit server
2559	0.72	0.72	0.72
2560	0.69	0.69	0.76
5145	0.69	0.67	0.64
2327	0.66	0.59	0.60
2557	0.84	0.50	0.86

Now, in Table 1 (page 13) we newly added the results of gmfit with gmconvert -maxsize 64 option in addition to the results using the default setting of the gmfit program which we originally provided.

(6) In page 14, the authors wrote “Comparing VESPER and CC, we see that VESPER had more cases with a smaller RMSD, indicating that the DOT score performs better than scoring with CC”, but, I do not think so. I feel Table 2 shows VES and CC generate very similar RMSDs, not indicating DOT is better than CC. In order to convince suspicious readers, the authors have to perform a rigorous statistical test if they want to say DOT is better than CC.

In Table 2, it is true that occasionally CC has a smaller RMSD than VES (VESPER with the DOT score), but overall VES has a smaller RMSD for more cases. When applied the t-test to evaluate the significance of the difference between VES and CC for the 15 map pairs, the p-value was 0.0080408, indicating that the better performance by VES is significant at the p-value level of 0.05. We added this to page 14-15.

(7) In Table 2, the authors should add RMSD values calculated by fitting of atomic model by MMalign. It is because if the two conformations are more different, the fitting of corresponding maps becomes more difficult. RMSD of MMalign should be ideal minimum value for fitting.

As suggested, we added a column “RMSD (Å)” to provide RMSD values computed by MMalign. (Note that the RMSD values of alignments computed by VESPER, CC, gmfit and fitmap are deviation of the alignment from the golden-standard alignment by MMalign. Thus, the MMalign RMSD can be larger than RMSD by the alignment methods compared.)

A large RMSD value, 10.44Å for 4a0v and 4a0w (TriC) is due to the conformational difference of the two structures. 4a0v (Green) and 4a0w(cyan) have open and close forms, respectively.

(8) In Table 2, some complexes have symmetries; TRPML ch has C4, ClpB-ClpP has C7 symmetry. Did the author care their symmetries when they calculated RMSD? The reason why I asked about question is, fitmap often returns very large RMSD (52-97 angstrom). Is it due to the symmetry problem?

Yes, symmetry was considered in RMSD calculation. We added this clarification in the table caption of Table 2. But still some alignment had a large RMSD. For example, fitmap had an RMSD of 70.2 Å for aligning EMD-8881 (PDB: 5wpq) and EMD-8764 (PDB: 5w3s). In the figure below, green, fitted 5w3s by fitmap and cyan is 5wpq. As you see in the figure, the structures are flipped against each other.

(9) In Table 2, the pair EMD-9758 and EMD-9778 is not good for bench mark, because the corresponding atomic models 6j0a and 6j45 only cover very small parts of the maps, they cannot be used as the correct standard for the fitting. The authors should choose another map pair “randomly” again.

Thank you for pointing it out. We removed the pair and replaced with EMD-4547 (PDB: 6qg5) and EMD-4548 (PDB: 6qg6) in Table 2.

(10) In the “Partial map alignment accuracy” section, a density map for each chain of atomic model was manually generated with the help of “Zone” function of UCSF Chimera. I think this test is not well-designed due to following two reasons. First, such a small density map is rarely obtained by current single particle analysis. Second, because the benchmark data was generated by the authors’ hands, other researchers cannot reproduce these maps to evaluate their upcoming new methods. To rescue these problems, I recommend to perform to fit a chain-based atomic model on the map, instead of a chain-based density map on the map. Because an atomic model is converted to the map for fitting (like colores in SITUS), the authors can do it by subtle modification of their source code. If the authors really do not want to perform model-to-map alignment, the authors have to open their hand-making bench-mark data in their WEB site or somewhere.

Thank you for the suggestion. We found that the suggestion is very reasonable. We followed the idea and now completely replaced the data in Figure 5 (Figure 4 of the original submission) and associated Supplemental Figures to new results of fitting chain-based atomic models to the maps. The text describing Figure 5 was completely written in page 20-21. The results show that VESPER with the DOT score performed in general better than the other methods.

The dataset we used for the atomic model fitting is provided for readers to use at http://kiharalab.org/vesper_data.

(11) In Eq.2, the authors’ kernel is defined as $\exp(-1.5 |p/\sigma|^2)$. The standard Gaussian function is defined as $\exp(-0.5 |p/\sigma|^2)$. Do they have a special reason to employ 1.5 instead of 0.5 ?

We used 1.5 because it is used in the SITUS package.

(12) I guess two parameters are important for representing maps: bandwidth sigma (=8 angstrom) and voxel spacing (= 7 angstrom). However, the authors employed smaller voxel spacing (1, 3, and 5 angstrom) in Table 2 (Global alignment). When the voxel spacing =1, 3, 5 angstrom, did they employ smaller bandwidth less than 8 angstrom? How much was voxel spacing for the “Partial map alignment accuracy” calculation?

The bandwidth is kept as a constant in all calculations. We added this clarification at the Equation 2.

For “Partial map alignment accuracy”, which now fits individual subunit structures into maps, we used 4 voxel spacing, 7, 5, 3, 1 angstrom as used in Table 2.

(13) For a typical map, how many number of mean points are generated? Please add number of mean points for the case of EMD-8409 and EMD-8724 in the legend of Fig.1.

A unit vector is calculated for each voxel in the map, not only for mean points. We added this clarification in page 6. The number of points for EMD-8409 and EMD-8724 were 678 and 2441, respectively. The numbers were added to the legend of Figure 1.

Responses to Comments by Reviewer #2:

In this manuscript, the authors describe a new method to quickly align EM maps by first generating a set of gradient vectors, and aligning vectors in two maps at a time by conversion to Fourier space.

Overall the paper is written well, the method is described clearly, and the results are well-organized. It is a novel and innovative method than others in the field, mainly through the use of gradient vectors for alignment purposes. However, gradients are known to be easily disrupted by noise, which is typically the case in cryoEM maps, and it is not clear if this was thoroughly addressed.

I think such a paper is timely in the field and the method could be useful, however I do have concerns with the limited accuracy at high resolutions. The advantage to other methods, e.g. fitmap, which can be tuned to have a wider search, or exhaustive search methods, other than speed alone, seems limited. Some parts may also need a bit more clarification as detailed below.

Introduction

- It is stated that “an accurate algorithm for aligning, comparing, and searching maps would substantially increase the value of the accumulated maps in EMDB”. It may not be too clear how this is so, and some points to elucidate some use cases could be interesting to add. It is typically more common to match an atomic structure (e.g. from the PDB) to a map, so as to search new

maps for known molecular components. Perhaps this may be a point where such a method would be useful, i.e. to match an existing map with known protein components to the new map. If this method only works between maps, a model could be first turned into a map and used for search, making it more useful.

I agree that the sentence was not well written. We now have written the sentence as follows:

“To interpret the structural information contained in EM density maps, alignment of maps is an essential step. Alignment is involved in many important procedures for analyzing EM maps, including rigid-body structure fitting of atomic models⁵, comparison of maps to identify differences in maps during a structure modeling process⁶ or to understand similarities and differences of different functional states⁷⁻⁹, and EM map database search^{10,11}. “

Relevant works are cited in the sentence in Introduction. Related to this, to respond to Reviewer 1’s comment (10), we now completely replaced the partial map alignment results with data for atomic model rigid-body fitting.

- Several existing methods are mentioned, such as gmfit, fitmap. Some other commonly used ones like Situs, ADP_EM, phenix.place_model_in_map, and Foldhunter are not mentioned. These are a lot more thorough than fitmap and would offer a more challenging comparison. Some may only place models, but at least ADP_EM and Foldhunter should work with map-map as well. I would say such comparisons may be not be necessary to perform for publication because it would add extra work, but they would make the paper a lot more informative and potentially relevant – i.e. a very important question is: is this method as good or close to an exhaustive search, which even though computationally expensive, is the only way to make sure good potential fits are not missed?

We mentioned and references are now cited ADP_EM, Situs, and Foldhunter with several other related papers in Introduction.

We tried to run ADP_EM and Foldhunter. For both methods, the computational time needed was too costly for our computer environment to perform the global map search benchmark which we performed in Figure 2 and Table 1. ADP_EM takes 600-1000 seconds per pair, which would mean the database search we did would take 600-1000 * (645*645 map combinations)= 2889 CPU/days. Foldhunter took about 30 hours per pair; thus the database search benchmark would take 520031 CPU/days. ADP_EM also caused segmentation fault error for many map pairs and we could not retrieve results. Therefore, we could not compare against ADP_EM and Foldhunter.

Regarding the exhaustive search, VESPER (and CC) performs an exhaustive search but with a combination of angle and translational interval using FFT. We added a new phrase to clarify it in page 4.

Results

- The voxel spacing of 7Å seems extremely large. This map should have an original voxel spacing of ~1Å or so. Were other voxel spacing tried and how do they affect the results? This

likely means that maps are only matched at low resolutions, but what does this mean for high-resolution maps?

For map alignment, as shown in Table 2, we tried finer translation and rotation intervals and they performs better than 7 Å in many cases. But for global map database search (where the aim is to find other EM maps of the same macromolecules), 7 Å seemed to perform sufficiently well relative to other methods as shown in Table 1 and Figure 1. In Supplementary Figure 3, we have also tested other combinations of translations and angles for the global map search.

- Using the Z-score is a good way to go. This has been used before rather than pure match score to show that fits are good.

Thank you.

- Figure 2. It is not clear if the text on page 11 is a caption or the main text.

Thank you for pointing it out. We changed the font size of figure and table captions to 11 point, which is smaller than the main text (12 point).

- It is not too clear how the CC score is used. Sounds like it is still with the VESPER search, but in that case, it's not the VESPER search that is important, but rather the use of the DOT score. In that case, the statement 'VESPER had a better retrieval performance than CC' seems ambiguous.

That is correct. The search is done by FFT with VESPER, but CC used cross-correlation as a scoring function rather than the DOT score. We changed the phrase to "VESPER with the DOT score" or "VESPER (DOT)" throughout the manuscript.

- Comparison to other methods (gmfit, fitmap, 3DZD). The results are well presented and quite thorough. However the other methods could potentially perform just as well if they are given different parameters (e.g. to increase their search space). It is not too clear how the parameters were chosen, other than to make running times similar.

The parameter setting of the three methods are described in page 12. In principle we used parameter settings that are used as default in their program or web servers. For gmfit, we now used two settings, following the two web servers developed by the gmfit author, which use gmfit as the main computational engine.

- Partial map alignment accuracy: as stated, all methods performed rather bad at 5.0Å and higher (note: the phrase "relatively higher resolution of less than 5.0 Å" may be stated a bit better). Could a more thorough search with voxel spacing of less than 7Å in the VESPER search help?

Yes, voxel spacing of less than 7 helps. Note that to answer a comment by Reviewer 1, we have now completely replaced the partial map alignment accuracy data in Figure 5 (which corresponds to Figure 4 in the initial submission) to results of mapping individual atomic detailed structures to density maps. In Supplementary Figure 4 in Supplementary file, we showed results with spacing less than 7 Å, 1 Å, 3 Å and 5 Å. As shown in the figure and the text, using finer spacing makes alignment results better.

For alignment, smaller spacing shows better performance. On the other hand, for database search, 7 Å seems to be sufficient for accuracy.

- It is a bit discouraging that the method would fail at higher resolution, where more detail should allow for higher accuracy. This may come back to the score used rather than the method, and exhaustive search should be used. This seems to say that VESPER is not comparable to exhaustive search unfortunately, but if speed is needed rather than accuracy, it would have the advantage.

VESPER uses Fast Fourier Transform (FFT) to do an exhaustive search, but with angle and translation interval. A sentence in page 4 is revised for the clarification.

In our observation, high resolution EM maps are not necessarily easier to correctly align, in many cases rather opposite, because a map has more dense local points, which can trap the alignment in local minimum.

But note that now Figure 5 is totally replaced with the results of structure fitting, where simulated maps of individual chains were fitted to the entire EM map. As shown in Figure 5b, now VESPER (DOT) performed better for higher resolution maps.

Discussion

- Perhaps VESPER may not be described as an ‘accurate method’ as per the results above, especially at higher resolutions. Perhaps this can be stated more in a way to indicate that it has higher accuracy under certain speed requirements and for a given range of resolutions.

Phrases such as “accurate method” are removed from Discussion. I agree that VESPER is not the absolutely accurate method in the absolute sense. It is all relative and also depends on the conditions tested. To clarify, in Discussion we added the following sentences:

Overall, VESPER showed higher accuracy than existing methods in both global and local EM map search and alignment under a reasonable speed requirement and for a given range of resolutions under the parameter settings tested. Note that, in general, the optimal parameter setting for a method differs for each map and the purpose of the computation. Thus, a perfectly fair comparison is not possible and the comparison shown in this work is to characterize the performance of VESPER but not to rank the methods.

Methods

- It would be interesting if the parameters for the other method used were given and discussed

briefly. E.g. for the fitmap method, what was N? Fitmap uses random placements so different runs could give different results?

In fitmap, we used $N = 100$, which we thought reasonably balances the speed and results. parameters of fitmap and gmfit, 3DZD are explained in page 12.

Some final thoughts:

- the use of local gradients may mean that noise is amplified. How can this be avoided? Does the reduction to voxel size of 7\AA use appropriate smoothing or just binning? If smoothing is used, this may help with reducing the effect of noise.

Yes, I agree with your insight. In fact, in the process of computing the vector representation, a Gaussian smoothing is applied, which is Equation 2 in page 24. We added the following “The Gaussian kernel has an effect of reducing density noise.” to state the purpose of this Gaussian kernel.

- The use of a threshold would ideally be avoided. This means that if the user specifies a different threshold, then the method would have different results. Also, all maps could be considered even if a threshold is not given.

I agree in the ideal scenario. VESPER could take in the whole map as input by, for example, setting the contour cutoff level to 0. But this will certainly include many noise in the density map box, which will probably deteriorate the accuracy. Moreover, the computational time would also drastically increase because the size of the maps to be compared will be larger. Therefore, if we want to input the entire map, we will need to do thorough benchmark particularly for the use of the entire map. And most probably, we will need to change the settings and revise algorithms of VESPER, to maintain the reasonable accuracy and the speed which would be not a trivial change.

Review by Grigore Pintilie

Responses to Comments by Reviewer #3:

With the rapidly increasing number of cryoEM structures determined, tools for comparison of this data is very relevant and useful for the community. The proposed approach uses vector representation of volumes based on mean shift algorithm, used in a previously published method for protein structure modelling from EM volumes. The authors report improvement over other related approaches for retrieval of related structures from a dataset. Although the proposed tool looks promising, the strengths (and weaknesses) have to be demonstrated better with more examples and benchmarks. The computation time is a limitation as well for database searches. I

have a few major concerns to do with demonstration of strengths/weaknesses of the proposed approach and examples to help user interpret results.

I would like to mention that in this revision, we added two new major data, a comparison of the DOT score (the vector-matching score used in VESPER) against 18 other existing scores (Supplementary Table 2). Also, the local map alignment data is now completely replaced with fitting of atomic-detailed structures to the density maps, responding to other reviewer, which highlight the usefulness of VESPER in structure-fitting purposes. I believe these two major new data and other revised data better characterize the performance of VESPER.

1) The computational time is a very important factor for database searches. More accurate search retrieval requires finer rotational sampling (10 \circ , Table 2, Table S2). This table is important from a user's point of view and has to be moved to the main text. The computational time can perhaps be reduced with volume pre-processing into vector representations? Strategies have to be discussed.

We have moved the Supplementary Table to the main text as Table 3. To further speed up, some pre-filtering may be a good approach, such as removing maps from the dataset that has significantly different overall volume from the query map, or some classes of maps, e.g. virus entries, or ribosome entries, can be removed if the user is not interested in them. We added this discussion at page 25.

2) The approach is compared against other related methods for volume comparison and alignment. However, the parameters used for other methods have to be defined clearly. For example:

a) For CC: Were the volumes resampled into 7Å voxels?, and maps contoured before score calculation? Was the calculation limited to volume of overlap between the two maps? CC is quite sensitive to these factors.

Yes, voxel spacing in CC is resampled to 7 Å (or the user's setting) and maps are contoured before calculation. Both CC and the DOT score were computed for the overlap between the two maps. This is because the DOT score need vector gradient representation, which can be computed only for regions with density, and also because in general, CC computed for the overlap region performed better in our experience.

We clarified this in page 11.

b) how was the number of gaussians decided for running gmfit? Omokage server retrieves slightly different results to gmfit. For example: search with EMD-2558 seems to retrieve 2557 as top hit, and 2556 and 2560, before EMD-5145 and EMD-2327 (Fig 2h).

The default setting of the standalone program of gmfit is different from the server. In the server, the gmconvert program, which converts an EM map to a Gaussian mixture model, uses an option -maxsize24. Since we used the standalone gmfit, the result was different from the server.

The results of running “gmfit with (gmconvert -maxsize 64)” on our local machine was consistent with the Omokage server:

Query:
2558

EMID	gmfit with (gmconvert -maxsize 64)	gmfit	Gmfit server
2559	0.72	0.72	0.72
2560	0.69	0.69	0.76
5145	0.69	0.67	0.64
2327	0.66	0.59	0.60
2557	0.84	0.50	0.86
2556	0.77	0.50	0.74

In Table 1, we now newly added results for gmfit with -maxsize24 option. The results with the option was slightly better than the original data we showed for gmfit.

c) For fitmap, were the maps pre-processed and contoured? Which metric was used for alignment?

Yes, the maps are contoured for fitmap. The Corr score was used for alignment. We added this in page 11-12.

3) Comparison of maps at different resolutions: In the tests for global and local accuracy, the alignments were tested within resolution categories. Is there a reason for not testing across the resolution ranges? This has to be demonstrated and discussed. Also for the case of comparison of a range of different conformations, it is useful to demonstrate this with a few examples.

This is a good suggestion. In Table 2, we added six comparisons with maps from different resolution ranges (the bottom rows of Table 2, labeled as cross-resolution.). Map pairs are selected so that the resolution ranges from relatively higher resolution (~3.5 Å) to a lower resolution (~15 Å).

4) For the global searches, I would be interested in knowing how the performance is when multiple groups of a single class are included in the search dataset, whether the partial alignments are ranked lower to full alignments?

It is an interesting question. We examined with the new Figure 4 (page 19). We used the data for the partial search, because the analysis use map classes, where a single class may include multiple groups. As shown in Figure 4, the results depend on the maps. About half of the cases, the top hit was from the same group and the other half cases retrieved a map from the same class as the top. Thus, sometimes a partial map alignment had a higher rank than full map alignment, but not always.

5) For VESPER, which parameters need user optimization? Do you recommend a Z-score cut-off for searches and alignments?

Thank you for the good question. To seek for the proper Z-score cutoff, we plotted Z-score distributions for global and partial map search, and global and partial map alignment in Supplementary Figure 5. From the plots, we think 10 would be a proper Z-score cutoff . We added the discussion in page 27.

6) Figure 1 has to be improved. Would be more intuitive to indicate positive DOT score positions with blue and negative with red. Also the atomic model segments of V-ATPase and Vo can be shown in the context of full model superposition.

We have revised Figure 1. We switched the blue and red dots and put the figure with full mode. Also, Figure 3 was revised to switch blue and red dots.

REVIEWER COMMENTS

Reviewer #1 (Remarks to the Author):

I appreciate the authors' effort to improve the manuscript according to the reviewers' requests. But I have further small requests to clarify the authors' algorithm and computation speed.

(1) For the searching algorithm, the authors just mentioned FFT. When I read the first draft, I thought the authors employed the standard "correlation" technique popularly used in protein-protein docking (Katchalski-Katzir et al. PNAS, 1992, 89:2195; Chacón and Wrigger, J Mol Biol. 2002 317:375). But I now recognize the standard FFT correlation does not work because the authors used the 3D vector field (each voxel has a unit vector), instead of 3D map, and employed the dot product score of the unit vector. The authors have to clearly describe how the authors use FFT to calculate the optimal translational vector for the dot product score between two 3D vector fields. I guess the authors prepared three types of 3Dmaps (for X, Y and Z components of the unit vectors), and calculated the correlation of these three maps, and got the sum of the three to get dot product values. It means the authors' algorithm requires three times longer computation times than the standard algorithm with cross correlation or Laplacian. If my guess is correct, the authors should add the sentences noting that their algorithm requires three times larger computation, but may have a higher accuracy than the standard algorithm.

(2) Table 4 summarized generating map representation for the query and the dataset for the retrieval; they are for FFT for the 820 dataset maps and rotated query maps. The authors did not show a searching computation time for VESPER, whereas they showed those for gmfit, fitmap and 3DZD). FFT of the 820 maps have to be executed one time, but FFT of rotated query maps have to be executed for each different query map. A searching time should include costs for FFT of rotated query maps and correlation calculation in Fourier space. And, I also want to know computation time for a pairwise 3D superimposition.

(3) Is a computation cost of VESPER is similar to that of Cross Correlation and Laplacian except mean shift process? If it is different, I want to know computation costs for Cross Correlation in Table 4. If my guess is correct, the cost of VESPER is at least three times of that of CC.

(4) To show the DOT score had more cases with a smaller RMSD than CC, the authors did the t-test and wrote "The difference of VESPER with the DOT score and CC for the 15 maps has a p-value of 0.008 when tested with on-sided paired test"(page 14). The authors should show more details about the t-test. At least, the authors have to show the averaged RMSD values used for the t-test, and the details of the dataset (grid spacing and angle spacing, and resolution range).

Reviewer #2 (Remarks to the Author):

The authors have addressed my questions and clarifications very well. I recommend publication.

Reviewer #3 (Remarks to the Author):

Thanks for the response and I really appreciate the additional work required to address reviewers' comments.

Some of my comments are partly addressed in this revised version: I believe some of them are quite important from the users' point of view and addressing this will help interpreting the advantages/limitations of the proposed method.

I also have a few concerns on the additional work included in this revised version.

I recommend that the authors please address these points/concerns below:

1) Following my question on the parameters used for methods used for comparison with VESPER, please include the number of random search positions used for chimera fitmap.

2) Following my question on the comparison of maps at different resolutions. Thanks for including examples of cross-resolution alignment in Table 2.

a) In the case of alignment between EMD-5001 and EMD-2001 which are GroEL in different conformations. Based on CCC the best alignment corresponds to a rmsd of >80. Are these cases of partial superpositions where only one ring of GroEL is aligned. As CC is computed only in the overlap between the two maps, the results might be misleading as it is a case of partial alignment and not incorrect (as rmsd reflects). If this is the case, please include the reasoning in the text for this case and also for others. It is also useful to show/mention the nature of alignments (extent of overlap between the maps) generated by VESPER in the sampling for global alignments.

b) Also for the two cases where fitmap finds poses closer to reference: if my understanding is correct, fitmap optimizes CCC for alignment and hence the best fitmap alignment is selected by CCC score. Is the reason CCC ranks a VESPER pose very different from reference (in terms of rmsd), again due to partial alignments? A supplementary figure showing some alignments of these cases could help.

Please include some discussion in the text – useful from the users point of view as CCC is the most used score as you say in the text, and is useful for readers to understand why/when it fails.

3) Following my question on the parameters requiring user optimization for VESPER run, the authors say the general recommendation of voxel spacing is 7Å and 30o angular sampling. Does the choice of contour level also affect the results, any comments on this would be useful.

4) Additional work corresponding to Suppl table 2 on the comparison of different scores, and in general for different tests that are part of this paper:

a) I believe rmsd is quite sensitive to the size of the structure and use of an absolute rmsd value as a cutoff to determine correct alignments is perhaps not the best thing to do. For large structures a few degree rotation can result in high rmsd values that could be misunderstood as pointing to an incorrect alignment. Also at medium/low resolutions, one often tends to find multiple alignments with the same/similar score and it is hard to say one alignment is the 'best'. Please include some discussion which will help readers interpret the results better.

b) The absolute value of Z-scores depends on the shape of the score distribution and the number of near-optimal poses and incorrect poses in the distribution. In a large database search, the number of incorrect poses dominate the distribution and the resulting Z-scores of good alignment(s) is expected to be high.

c) To understand how the DOT Z-scores vary with rmsd, please include a supplementary figure showing score vs rmsd for a few cases from each category discussed in Suppl Table 2. Also include plots for some of the other scores compared here. I hope this doesn't require much additional work as the results are already there. The csv files for download don't seem to have a way to link the model ids to compare rmsd/ALCPS vs scores. From a quick look at Joseph et al and the results here, the values of the local scores (e.g. SMOC, LMI) are lower than the global (CCC,MI) scores. I would expect the local scores calculated in the overlap region to be higher in terms of range of values. Please ensure correct scores are used.

d) Suppl table 2: Please highlight all best scores in different categories, some of the 'best' values are not highlighted. Also, in this table please include the difference in (rotation,translation) from the reference, for the best scoring pose.

5) Additional work included corresponding to Fig.5: How many top scoring VESPER alignments were considered for finding best fitting chains or the best scoring alignment always correspond to lowest rmsd with reference?

Responses to Comments by Reviewer #1:

I appreciate the authors' effort to improve the manuscript according to the reviewers' requests. But I have further small requests to clarify the authors' algorithm and computation speed.

(1) For the searching algorithm, the authors just mentioned FFT. When I read the first draft, I thought the authors employed the standard "correlation" technique popularly used in protein-protein docking (Katchalski-Katzir et al. PNAS, 1992, 89:2195; Chacón and Wrigger, J Mol Biol. 2002 317:375). But I now recognize the standard FFT correlation does not work because the authors used the 3D vector field (each voxel has a unit vector), instead of 3D map, and employed the dot product score of the unit vector. The authors have to clearly describe how the authors use FFT to calculate the optimal translational vector for the dot product score between two 3D vector fields. I guess the authors prepared three types of 3Dmaps (for X, Y and Z components of the unit vectors), and calculated the correlation of these three maps, and got the sum of the three to get dot product values. It means the authors' algorithm requires three times longer computation times than the standard algorithm with cross correlation or Laplacian. If my guess is correct, the authors should add the sentences noting that their algorithm requires three times larger computation, but may have a higher accuracy than the standard algorithm.

Yes, that is correct. The FFT part of VESPER requires three times more computation time than CC. We added a new sentence to explain this on page 7.

(2) Table 4 summarized generating map representation for the query and the dataset for the retrieval; they are for FFT for the 820 dataset maps and rotated query maps. The authors did not show a searching computation time for VESPER, whereas they showed those for gmfit, fitmap and 3DZD). FFT of the 820 maps have to be executed one time, but FFT of rotated query maps have to be executed for each different query map. A searching time should include costs for FFT of rotated query maps and correlation

calculation in Fourier space. And, I also want to know computation time for a pairwise 3D superimposition.

We have revised Table 4 extensively. We provided computational time for the DOT score at the upper half of the table, and put the time for cross-correlation (CC) at the lower half. As suggested, we have split the computational time for preparing files, i.e. computing the vector representation (for the DOT score), and applying the Gaussian kernel for CC.

The time shown is the total search time for a query map against the 410 maps. The number of maps in the global map search dataset is 410. In the parentheses, as requested, we provided times needed for the pairwise comparison.

Apparently, using CC takes less time than using DOT score, but it is not one-third, because file access and other overhead operations also take time when using CC.

(3) Is a computation cost of VESPER is similar to that of Cross Correlation and Laplacian except mean shift process? If it is different, I want to know computation costs for Cross Correlation in Table 4. If my guess is correct, the cost of VESPER is at least three times of that of CC.

In Table 4, we also added the measured computational time for CC. As mentioned in the previous question, FFT part of the DOT score with VESPER has 3 times larger time complexity than using CC. But practically, as shown in Table 4, the overall computational time for CC is not exactly reduced to 1/3, because other operations, including file access (Input and Output) basically takes the same, relatively a large amount of time for CC, too.

(4) To show the DOT score had more cases with a smaller RMSD than CC, the authors did the t-test and wrote “The difference of VESPER with the DOT score and CC for the 15 maps has a p-value of 0.008 when tested with on-sided paired test”(page 14). The authors should show more details about the t-test. At least, the authors have to show the averaged RMSD values used for the t-test, and the details of the dataset (grid spacing and angle spacing, and resolution range).

We added two new rows at the bottom of Table 2, the averaged RMSD values and the averages of RMSD differences between VESPER and CC. The t-test was performed for the distribution of RMSD differences between VESPER and CC for the 15 map comparisons in Table 2 (i.e. $15 \times 4 = 60$ values). We rephrased it to “The difference of RMSD values between VESPER with the DOT score and CC for the 15 maps..”.

Responses to Comments by Reviewer #2:

The authors have addressed my questions and clarifications very well. I recommend publication.

Thank you very much.

Responses to Comments by Reviewer #3:

Thanks for the response and I really appreciate the additional work required to address reviewers' comments.

Some of my comments are partly addressed in this revised version: I believe some of them are quite important from the users' point of view and addressing this will help interpreting the advantages/limitations of the proposed method.

I also have a few concerns on the additional work included in this revised version.

I recommend that the authors please address these points/concerns below:

1) Following my question on the parameters used for methods used for comparison with VESPER, please include the number of random search positions used for chimera fitmap.

It is 100. On page 12, we stated that “In fitmap, an input map was contoured, and the number of initial placements was set to 100.”

2) Following my question on the comparison of maps at different resolutions. Thanks

for including examples of cross-resolution alignment in Table 2.

a) In the case of alignment between EMD-5001 and EMD-2001 which are GroEL in different conformations. Based on CCC the best alignment corresponds to a rmsd of >80. Are these cases of partial superpositions where only one ring of GroEL is aligned. As CC is computed only in the overlap between the two maps, the results might be misleading as it is a case of partial alignment and not incorrect (as rmsd reflects). If this is the case, please include the reasoning in the text for this case and also for others. It is also useful to show/mention the nature of alignments (extent of overlap between the maps) generated by VESPER in the sampling for global alignments.

Actually, not only CC but also the DOT score computed the score in the overlapped region between two maps. So, that part is the same for CC and DOT score. This has been stated on page 11 (the sentence is highlighted in yellow).

To clarify how those aligned structures with large RMSD values look like, we made new Supplementary Figure 2, which shows all the alignments with an RMSD of over 40 Angstroms by CC as well as by fitmap in Table 2. Supplementary Figure 2 is referred to on page 15.

As shown in the figures, the overlapped region by two maps in these alignments was not small. The reason of the different performance of CC and DOT score is not due to the overlapped region. We think there is a fundamental difference between scoring with the DOT score and CC. We discussed it in Discussion (page 23). I explained it in the next page of this letter.

Related to this question and the subsequent questions about CC, we added a paragraph to mention about the conventional equation we used to compute CC, because we noticed that it was not clearly mentioned in the previous version of the manuscript.

b) Also for the two cases where fitmap finds poses closer to reference: if my understanding is correct, fitmap optimizes CCC for alignment and hence the best fitmap alignment is selected by CCC score. Is the reason CCC ranks a VESPER pose very different from reference (in terms of rmsd), again due to partial alignments? A

supplementary figure showing some alignments of these cases could help.

As mentioned in the previous question, we now showed the alignments with large RMSD values by fitmap in the new Supplementary Figure 2. As can be seen in the figure, map overlap in failed alignments by fitmap is not small. A difference of fitmap is that it performs local optimization starting from randomly selected alignment positions, which we set to 100 positions. So fitmap does not perform a conformational search over the entire rotation and shifting space and often does not find global optimal alignment. The fitmap method has been described on page 3. We commented on the nature of local optimization by fitmap on page 15.

Please include some discussion in the text – useful from the users point of view as CCC is the most used score as you say in the text, and is useful for readers to understand why/when it fails.

Thank you, we agree that this is an important point to discuss. I'll explain here what we think. As you see in the new Supplementary Figure 2, overlaps of two maps in CC alignments were not very small. Actually, the DOT score also only considers the overlapped region in the alignment, in the same way as CC does.

But the vector comparison by the DOT score in VESPER can often make different alignment selection from what CC prefers:

In CC, due to the equation of CC (the equation of CC is provided in the Methods section), aligned positions (position i 's in the numerator of the equation) from two maps that have both large positive normalized density values contribute relatively largely to the entire CC. On the other hand, the aligned positions close to the average density value (thus, have a normalized value of around 0) have less contribution because the multiplied value for that position will be around 0. Thus, using CC, a map alignment will have a large CC value (and will be selected) if regions with high density are aligned.

In contrast, DOT score is computed by aligning unit vectors that point to neighboring denser points. Since the vectors are unit vectors (equal lengths), regardless of the row density values, each aligned position from two maps basically contributes

equally to the entire score of the two maps. This can result in a difference in the scores given by DOT score and CC. And it turned out that the DOT score performed better in more cases than CC.

We added this discussion in Discussion (page 15).

3) Following my question on the parameters requiring user optimization for VESPER run, the authors say the general recommendation of voxel spacing is 7Å and 30o angular sampling. Does the choice of contour level also affect the results, any comments on this would be useful.

Yes, the contour level would affect VESPER's performance since we want to include density regions that correspond to molecular structure but exclude density of outside noise. In this work, we used the author-recommended contour level stated in each EMDB entry to standardize the evaluation process, which we now stated on page 6 and 26.

4) Additional work corresponding to Suppl table 2 on the comparison of different scores, and in general for different tests that are part of this paper:

a) I believe rmsd is quite sensitive to the size of the structure and use of an absolute rmsd value as a cutoff to determine correct alignments is perhaps not the best thing to do. For large structures a few degree rotation can result in high rmsd values that could be misunderstood as pointing to an incorrect alignment. Also at medium/low resolutions, one often tends to find multiple alignments with the same/similar score and it is hard to say one alignment is the 'best'. Please include some discussion which will help readers interpret the results better.

I agree with the reviewer that RMSD is not a perfect metric and can be too sensitive, like the cases discussed by the reviewer. In the Suppl Table 2, we have also provided log(ALCPS), which was used in the paper by Joseph et al and kindly referred by the reviewer. The reason why we also provided RMSD was, as mentioned in the previous revision letter, because we observed cases where log(ALCPS) is quite permissive, and visually very different alignments can often achieve very good log(ALCPS) score (this

happens for log(ALCPS) because it multiplies an angle deviation and a translational deviation, so if one of the deviations, e.g. angle, is very small, a good log(ALCPS) score is possible with very large translational deviation).

Since we realized that no score is perfect, in Suppl Table 2, we provided both log (ALCPS) and RMSD, and moreover, in five different flavors, AUC, accuracy, and z-score with two cutoffs. To clarify, on page 27 and 28, we stated why we provided both log(ALCPS) and RMSD, in different flavors. The sentences start with:

These scores evaluate different important aspects of the scores. log(ALCPS) concerns deviations of translational and rotational shifts of an alignment where as RMSD measures an average deviation at each amino acid residue point. ...

b) The absolute value of Z-scores depends on the shape of the score distribution and the number of near-optimal poses and incorrect poses in the distribution. In a large database search, the number of incorrect poses dominate the distribution and the resulting Z-scores of good alignment(s) is expected to be high.

Yes, that is correct. A large absolute value of a z-score means a better and a distinct score relative to the background distribution. We added the explanation of z-score on page 27.

c) To understand how the DOT Z-scores vary with rmsd, please include a supplementary figure showing score vs rmsd for a few cases from each category discussed in Suppl Table 2. Also include plots for some of the other scores compared here. I hope this doesn't require much additional work as the results are already there. The csv files for download don't seem to have a way to link the model ids to compare rmsd/ALCPS vs scores. From a quick look at Joseph et al and the results here, the values of the local scores (e.g. SMOC, LMI) are lower than the global (CCC,MI) scores. I would expect the local scores calculated in the overlap region to be higher in terms of range of values. Please ensure correct scores are used.

We added the new plot, Supplementary Figure 7, which compares DOT Z-score with rmsd. In addition to DOT Z-score, we compared rmsd with four other scores: Cross

correlation-coefficient (CCC), Chamfer surface distance score on all points at an iso-contour level (CDA), Normal vector score on all points at an iso-contour level (NVA), and Overlap (OVR). We selected CCC, CDA, NVA, and OVR because they are different types of scores among the 18 scores used in Supplementary Table 2. CCC is correlation-based, CDA is surface-based, NVA is normal vector-based, and overlap is volume-based. For each comparison, we have chosen 4 map pairs from each of the three protein categories, Virus, Ribosome and Other. As shown in Supplementary Figure 7, for most of the cases, the scores correlate well with RMSD, i.e. a lower RMSD alignment gives significant scores. For map alignment for virus maps, depending on aligned maps, some scores, including DOT Z-score, have a weaker correlation but the correlation still exists. The new figure is referred to on page 28.

d) Suppl table 2: Please highlight all best scores in different categories, some of the 'best' values are not highlighted. Also, in this table please include the difference in (rotation, translation) from the reference, for the best scoring pose.

We revised Suppl. Table 2. We highlighted the best value for each column (i.e. each metric) in the table. Also, we added a new column, "rot. Trans of best scoring alignment" in the first sheet of the Excel file, where the rotation and the translation of the alignment selected by the score is provided. Since this table provides one value for a score for each map category (others, ribosome, virus), the rotation and translation values provided are the average of all the alignments in the category. (note that we did not put the data in the second sheet, because the results are the same as what we put in the first sheet).

5) Additional work included corresponding to Fig.5: How many top scoring VESPER alignments were considered for finding best fitting chains or the best scoring alignment always correspond to lowest rmsd with reference?

We considered the alignment that has the highest score. We did not use RMSD values because we want to evaluate the performance of how well the scores select alignments.

REVIEWER COMMENTS

Reviewer #1 (Remarks to the Author):

The authors have addressed my questions and requests well. I recommend publication.

Reviewer #3 (Remarks to the Author):

Authors have tried to address the raised concerns but I recommend a few minor changes to fully address some of the earlier comments which I think is useful to interpret the strengths/weaknesses and the results of the proposed method (and score). I will be able to recommend publication of this work once the following minor changes are included.

1) On my earlier comment on partial alignments: The new supplementary fig 2 is useful to understand the quality of alignments from the approach, thanks for clarifying the score calculations. The effect of partial alignments is a known issue and not many scores tackle this. Regarding fig S2:

Both CCC and DOT scores are calculated in the volume of overlap.

- DOT score is a cumulative dot product of matched vectors, so larger the overlap and number of matched vectors, larger the DOT score? or partial alignments are scored lower? I would appreciate a discussion on this to elaborate further on the comparison between CCC and DOT.

2) Supplementary figure 2:

In the case of best alignment of emd-2559 and emd-2557 by fitmap,

- the equivalent structure for emd-2557 is not clear in the fitmap alignment! The green and pink structures in MMalign and fitmap alignments, don't seem to match to me.

3) The new section added to discussion :

“This implementation of the DOT score often results in different preference of map alignments as compared with CC. In CC, aligned positions from two maps have a greater effect on the overall score if the two positions have higher density values relative to the average. Oppositely, contribution of aligned positions would be smaller if their density values are around the average. On the other hand, in the DOT score, the contribution of each pair of aligned positions are essentially the same because the vectors are normalized to the same length of 1.0 (unit vectors).”

The whole distribution, its shape and standard deviation contributes to the CCC value. The normalisation with respect to the spread of the distribution is I think what the authors want to highlight. On the other hand, the local gradient used for calculating DOT score is affected by the local variations, although the Gaussian kernel helps reduce this to a certain extent.

- Please rephrase this paragraph as this point is not very clear.

4) Following my earlier comments on Supplementary table 2:

- The AUC and Z-score values are not displayed in this version of the table. (in Excel) Could authors please fix this.

- Authors state that "Other scores that showed the top performance in terms of some metrics include a normal vector-based score (NVA in the table) and the cross correlation (CCC)". OVR is also highlighted by authors for best performance in the table based on some metrics. Therefore, please include OVR as well in the statement above.

- About my earlier question on the values of the scores: For the example emd-1056 vs emd-1895, using author recommended contours and map transformation used by the authors (http://kiharalab.org/vesper_data), I get the following results for the scores using the CCP-EM software:

Alignment 1:

Percent overlap: 0.547861721131
Local correlation score: 0.12465485
Local Mutual information score: 0.017558647512459835
Correlation score: 0.31208223
Normalized Mutual information score: 1.0378895432997726
Mutual information score: 0.12156281453595286

Alignment 2:

Percent overlap: 0.557678740299
Local correlation score: 0.14930685
Local Mutual information score: 0.023040487618832373
Correlation score: 0.3154713
Normalized Mutual information score: 1.0375545946863771
Mutual information score: 0.12269763273865131

Alignment 3:

Percent overlap: 0.518875996337
Local correlation score: 0.095742844
Local Mutual information score: 0.013273485104096316
Correlation score: 0.27708402
Normalized Mutual information score: 1.0368238603493334
Mutual information score: 0.11915735615494638

The LMI_OV and SCCC_OV scores in the output are :

ccc_ov 0.12 0.15 0.07
mi_ov 0.02 0.02 0.01

The ranks of these alignments appear different to what is mentioned in http://kiharalab.org/vesper_data. Please ensure correct scores and ranks are listed and cited in the article.

Response to Comment by Reviewer #1:

The authors have addressed my questions and requests well. I recommend publication.

Thank you.

Response to Comment by Reviewer #3:

1) On my earlier comment on partial alignments: The new supplementary fig 2 is useful to understand the quality of alignments from the approach, thanks for clarifying the score calculations. The effect of partial alignments is a known issue and not many scores tackle this. Regarding fig S2:

Both CCC and DOT scores are calculated in the volume of overlap.

- DOT score is a cumulative dot product of matched vectors, so larger the overlap and number of matched vectors, larger the DOT score? or partial alignments are scored lower? I would appreciate a discussion on this to elaborate further on the comparison between CCC and DOT.

> larger the overlap and number of matched vectors, larger the DOT score?

Yes. The DOT score is a cumulative dot product of matched vectors. Therefore, a larger overlap region tends to have a larger DOT score as long as a majority of matched vectors have a positive dot product value. We added this sentence at page 5.

2) Supplementary figure 2:

In the case of best alignment of emd-2559 and emd-2557 by fitmap,

- the equivalent structure for emd-2557 is not clear in the fitmap alignment! The green and pink structures in MMalign and fitmap alignments, don't seem to match to me.

That is correct. Fitmap did not align the two structures well (cyan and the magenta structures on the right). Supplementary Figure 2 panel a shows poor structure alignments by fitmap that had very large RMSD values. The RMSD of the fitmap alignment relative to the MMalign (on the left, green and cyan structures) is 44.31Å as indicated. To clarify the difference of the alignments by MMalign and fitmap, in the figure below we superimposed the two

alignments. 4d2u (EMD-2557) is shown in green; 4d2x (EMD-2259) aligned by MMalign is shown in cyan; and alignment by fitmap is shown in magenta:

These 2 structures, 4d2u and 4d2x are homo pentameric complex, but the conformation of the chains in the two structures are largely different. 4d2u (green) has a stretched conformation whereas 2d2x has a compact conformation, which made the alignment of the two complex structures difficult.

On the other hand, MMalign made a reasonable superimposition. As you see below, the C-terminal domain of each chain (residue 545-857. In the figure above, it corresponds to the upper part of the green and cyan structures), which did not make much conformational change, were aligned well by MMalign:

3) The new section added to discussion :

“This implementation of the DOT score often results in different preference of map alignments as compared with CC. In CC, aligned positions from two maps have a greater effect on the overall score if the two positions have higher density values relative to the average. Oppositely, contribution of aligned positions would be smaller if their density values are around the average. On the other hand, in the DOT score, the contribution of each pair of aligned positions are essentially the same because the vectors are normalized to the same length of 1.0 (unit vectors).”

The whole distribution, its shape and standard deviation contributes to the CCC value. The normalisation with respect to the spread of the distribution is I think what the authors want to highlight. On the other hand, the local gradient used for calculating DOT score is affected by the local variations, although the Gaussian kernel helps reduce this to a certain extent.

- Please rephrase this paragraph as this point is not very clear.

We wanted to mention the following difference in DOT and CC:

In DOT, since vectors are normalized to have the equal length, the dot product of the two aligned vectors ranges from -1 to 1. Thus, each dot product basically contribute equally to the the DOT score.

CC is different. Considering the numerator of the CC equation (in the manuscript it is Equation 3 in the Method section), positions that have a large absolute value of density tend to have a large contribution to the overall CC value, whereas positions that have an average density value may have negligible contribution because $u_i - \langle u \rangle$ will be close to 0 (zero). Therefore, different positions with different density values contribute differently to the overall CC value.

But as you now pointed out, this difference do not always make the DOT score advantageous. As you mentioned, the local gradient used for calculating DOT score is affected by the local variations. We added this point and rephrased the paragraph.

4) Following my earlier comments on Supplementary table 2:

- The AUC and Z-score values are not displayed in this version of the table. (in Excel) Could authors please fix this.

Supplementary Table 2 (in an Excel file), which is now renamed as Supplementary Data 2 following the regulation of the journal, do have AUC and Z-score values. In the first datasheet “RMSD” and in the second datasheet “log(ALCPS)”, AUC values are shown in column B, N, Z. They are labeled as “AUC” in row 2. Z-score values are shown in column D & E, P & Q, AB & AC. They are labeled as Z1 and Z10, which have following explanation in the page 3 of the Supplementary Information file: “Z-score 1 (Z1), the Z-score of the best alignment (i.e. lowest RMSD or log(ALCPS) score) computed from all the alignments; Z-score of top 10 models (Z10), the average Z-score of the 10 best (i.e. lowest RMSD or lowest log(ALCPS)) alignments.”.

If we are missing something, let us know.

- Authors state that “Other scores that showed the top performance in terms of some metrics include a normal vector-based score (NVA in the table) and the cross correlation (CCC)”. OVR is also highlighted by authors for best performance in the table based on some metrics. Therefore, please include OVR as well in the statement above.

We included OVR in page 12.

- About my earlier question on the values of the scores: For the example emd-1056 vs emd-1895, using author recommended contours and map transformation used by the authors (http://kiharalab.org/vesper_data), I get the following results for the scores using the CCP-EM software:

Alignment 1:

Percent overlap: 0.547861721131

Local correlation score: 0.12465485

Local Mutual information score: 0.017558647512459835

Correlation score: 0.31208223

Normalized Mutual information score: 1.0378895432997726

Mutual information score: 0.12156281453595286

Alignment 2:

Percent overlap: 0.557678740299

Local correlation score: 0.14930685

Local Mutual information score: 0.023040487618832373

Correlation score: 0.3154713

Normalized Mutual information score: 1.0375545946863771

Mutual information score: 0.12269763273865131

Alignment 3:

Percent overlap: 0.518875996337

Local correlation score: 0.095742844

Local Mutual information score: 0.013273485104096316

Correlation score: 0.27708402

Normalized Mutual information score: 1.0368238603493334

Mutual information score: 0.11915735615494638

The LMI_OV and SCCC_OV scores in the output are :

ccc_ov 0.12 0.15 0.07

mi_ov 0.02 0.02 0.01

The ranks of these alignments appear different to what is mentioned

in http://kiharalab.org/vesper_data. Please ensure correct scores and ranks are listed and cited in the article.

SCCC_OV and LMI_OV were computed by the TEMpy package. The details of SCC_OV and LMI_OV are described in

Joseph, Agnel Praveen, et al. "Improved metrics for comparing structures of macromolecular assemblies determined by 3D electron-microscopy." *Journal of structural biology* 199.1 (2017): 12-26.

In section 3.4, it says

<https://www.sciencedirect.com/science/article/pii/S1047847717300874?via%3Dihub>

“The combined score is the average of scaled and shifted OVR (OVRnorm) and SCCC/SMOC/LMI.”

SCCC_OV and LMI_OV in our results are not the same metrics with Local correlation and Local Mutual information score you computed by CCP-EM.

The Local correlation score in your computation corresponds to SCCC table in our dataset.

In our data table (http://kiharalab.org/vesper_data), alignments 1-3 of emd-1056 vs emd-1895 have following SCC values.

SCCC in (http://kiharalab.org/vesper_data)

0.13098798177130025 1

0.15427877102662163 2

0.10012614093240443 3

These values are consistent with the result from CCP-EM

Local correlation score (CCP-EM)

Alignment1 Local correlation score: 0.12465485

Alignment2 Local correlation score: 0.14930685

Alignment3 Local correlation score: 0.095742844

The values you computed as “Local Mutual information” equal to LMI in our data at http://kiharalab.org/vesper_data.. Our LMI results are:

0.01895570755 1

0.0259504318237 2

0.0164818763733 3

These values are close to the result you got by CCP-EM:

Local Mutual information score (CCP-EM)

Alignment1 Local Mutual information score: 0.017558647512459835

Alignment2 Local Mutual information score: 0.023040487618832373

Alignment3 Local Mutual information score: 0.013273485104096316